# Transformer Copilot: Learning from The Mistake Log in LLM Fine-tuning

**Jiaru Zou**[1], **Yikun Ban**[1†], **Zihao Li**[1], **Yunzhe Qi**[1], **Ruizhong Qiu**[1], **Ling Yang**[2†], **Jingrui He**[1†]

[1]University of Illinois Urbana-Champaign    [2]Princeton University

{jiaruz2, yikunb2, jingrui}@illinois.edu, ly1988@princeton.edu

 **Code:** https://github.com/jiaruzouu/TransformerCopilot

## Abstract

Large language models are typically adapted to downstream tasks through supervised fine-tuning on domain-specific data. While standard fine-tuning focuses on minimizing generation loss to optimize model parameters, we take a deeper step by retaining and leveraging the model's own learning signals, analogous to how human learners reflect on past mistakes to improve future performance. We first introduce the concept of *Mistake Log* to systematically track the model's learning behavior and recurring errors throughout fine-tuning. Treating the original transformer-based model as the Pilot, we correspondingly design a Copilot model to refine the Pilot's inference performance via logits rectification. We name the overall Pilot-Copilot framework the *Transformer Copilot*, which introduces (i) a novel Copilot model design, (ii) a joint training paradigm where the Copilot continuously learns from the evolving Mistake Log alongside the Pilot, and (iii) a fused inference paradigm where the Copilot rectifies the Pilot's logits for enhanced generation. We provide both theoretical and empirical analyses on our new learning framework. Experiments on 12 benchmarks spanning commonsense, arithmetic, and recommendation tasks demonstrate that Transformer Copilot consistently improves performance by up to 34.5%, while introducing marginal computational overhead to Pilot models and exhibiting strong scalability and transferability.

## 1 Introduction

Transformers, the foundation of modern large language models (LLMs), leverage attention and feedforward layers to compute logits for sequence generation [77]. Pre-trained on general-domain corpora, these models capture rich statistical patterns and exhibit strong generation capabilities [13, 80, 57]. On top of that, supervised fine-tuning (SFT) serves as a critical technique for adapting pre-trained LLMs to specific domains [40, 65, 80]. While SFT enables significant flexibility and task-specific optimization, the performance of fine-tuned LLMs during inference often remains suboptimal, exhibiting misalignment between training and testing stages [47, 79]. This gap arises from the model's inability to fully capture task-specific nuances or from overfitting to patterns within the training data, ultimately degrading its final performance [65, 56, 95, 53]. Without data-side interventions [52, 54, 27] or receiving external feedback [58, 72, 91], this paper aims to address a fundamental question: *Can we enhance the inference performance by retaining and leveraging the model's own learning signals in standard fine-tuning?*

To address this question, our core idea draws inspiration from a common strategy by human learners: maintaining a log to record mistakes during practice, reflecting, and using insights to improve performance in formal tests. Rather than merely memorizing these mistakes, proficient learners engage in reflective thinking—analyzing their internal cognitive states at the moment the errors

---

[†]Corresponding authors

39th Conference on Neural Information Processing Systems (NeurIPS 2025).

occurred, questioning how and why the mistakes were made. The reflective practice enables learners to identify recurring error patterns and approach uncertain problems with greater caution and awareness.

Motivated by this human reflection thinking mechanism [31], we propose the concept of **Mistake Log** tailored for LLMs' fine-tuning. At training stages, standard SFT primarily focuses on optimizing model parameters by minimizing the expected loss over fine-tuning datasets [80, 97]. We take a deeper step to systematically record the rich intermediate information within the model, including input data (Question), internal hidden state representations (Rationale), and token-level quantified errors (Mistakes), as Mistake Log components to track model's mistakes through its training trajectory.

Next, to fully exploit the Mistake Log, we propose the **Transformer Copilot** (abbreviated as T-Copilot), a novel Pilot-Copilot framework that enables error-aware refinement by learning from model-internal signals [11, 12, 34]. In addition to the original model (referred to as the Pilot), we introduce a Copilot model that captures and leverages the Pilot's Mistake Log throughout its learning trajectory, rectifying the Pilot's logits to improve final token-by-token generation. Overall, our learning framework offers advantages from three key perspectives: (i) **New Model Architecture Design:** We design the Copilot as a transduction neural network that learns recurring error patterns from the Mistake Log. A residual flow connection is then established between the Copilot and Pilot models, allowing the Copilot to assist the Pilot via token-level error correction during generation. (ii) **New Training Paradigm:** We redesign the SFT procedure by jointly training the Pilot and Copilot models in each round, enabling the Copilot to continuously learn from the evolving Mistake Log and adapt alongside the Pilot model. (iii) **New Inference Paradigm:** During next-token generation, we fuse the output logits from the Pilot and Copilot models into a unified probability distribution, enabling collaborative auto-regressive generation. In this way, T-Copilot fundamentally integrates an internalized reflection mechanism into standard SFT, enabling an adaptive and error-aware generation.

To demonstrate the efficacy of the T-Copilot, we provide detailed analyses from both theoretical and empirical perspectives. We incorporate T-Copilot into representative encoder-decoder and decoder-only Pilot models, and conduct extensive experiments across 12 tasks on commonsense, arithmetic, and real-world recommendation. T-Copilot improves the performance of Pilot by up to 34.5% while surpassing strong baselines with significantly fewer parameters. For example, integrating T-Copilot with Qwen2.5-7B outperforms Qwen2.5-14B using 4B fewer parameters. We further comprehensively study the efficiency, transferability, and scalability of T-Copilot, showing that T-Copilot brings marginal computational overhead to Pilot, scales well across different model types and sizes, and effectively transfers to new Pilot models for inference without additional training costs.

## 2 Definition of Mistake Log

### 2.1 Preliminary and Notations

Let $f^P(\cdot; \theta^P)$ denote the function computed by a standard Transformer model [77], parameterized by $\theta^P$. In our context, we refer to $f^P$ as the *Pilot* model. Suppose there are $T$ fine-tuning rounds. For each round $t \in [T]$, given an input sequence $X_t = (x_{t,1}, \ldots, x_{t,n})$ where $n$ is the maximum sequence length, the input is sampled from a data distribution $\mathcal{D} = \mathcal{D}_{\mathcal{X}, \mathcal{Y}}$ over input-output pairs. The Pilot model then generates an output sequence $\hat{Y}_t = (\hat{y}_{t,1}, \ldots, \hat{y}_{t,n})$ in an auto-regressive manner to approximate the target sequence $Y_t = (y_{t,1}, \ldots, y_{t,n})$, where $(X_t, Y_t) \sim \mathcal{D}$.

During $t$-th fine-tuning round, let $\widetilde{X}_t$ denote the input representation of $X_t$, defined as either the encoder output in an encoder-decoder Transformer or the output of the token and positional embedding layer in a decoder-only Transformer. In the forward pass through the residual stream of the model, let $L^P$ be the total number of decoder layers in the Pilot model. For each layer $l \in [L^P]$, we define $h_{t,i,l}(\widetilde{X}_t; \theta^P_{t-1})$ as the (decoder) hidden representations of the $i$-th token. After the final decoder layer, the Pilot model outputs logits over the vocabulary $V$, conditioned on the input $X_t$ and shifted target sequence $y_{t,<i}$. The resulting output probabilities for the $i$-th token are given by:

$$\hat{p}_{t,i} = \text{softmax}\left(f^P(X_t, y_{t,<i}; \theta^P_{t-1})\right). \tag{1}$$

We denote $p_{t,i}$ the ground-truth distribution over $V$ for the $i$-th token, which places full probability mass on the correct token $y_{t,i}$. The objective of training $f^P$ is to minimize the cross-entropy loss between the predicted and ground-truth tokens, formulated as:

$$\mathcal{L}^P_t = -\sum_{i=1}^{n} \log \hat{p}_{t,i}(y_{t,i} \mid X_t, y_{t,<i}). \tag{2}$$

## 2.2 The Mistake Log

Next, we define the *Mistake Log* in fine-tuning scenarios. As shown in Figure 1, the Mistake Log concludes three key components: the input representations (Questions), internal hidden states representations (Rationales), and the token-level error made by the model (Mistakes).

In each round $t \in [T]$, draw the sequence pair $(X_t, Y_t) \sim \mathcal{D}$. As defined in Section 2.1, we set $\widetilde{X}_t$ as the input representation component, as it provides contextual grounding for the Pilot model's specific input sequence. Inspired by prior works [25, 20, 14, 45], the intermediate states' hidden representations produced by Transformer blocks also encapsulate rich contextual and semantic information, reflecting the model's internal rationales. Therefore, we define $h_t(X_t; \theta_{t-1}^P)$ as the collection of these internal hidden representations for each token in round $t$:

$$h_t(\widetilde{X}_t; \theta_{t-1}^P) = \left\{ h_{t,i}(\widetilde{X}_t; \theta_{t-1}^P) \right\}_{i=1}^n, \quad \text{with } h_{t,i}(\widetilde{X}_t; \theta_{t-1}^P) = \left\{ h_{t,i,l}(\widetilde{X}_t; \theta_{t-1}^P) \right\}_{l=1}^{L^P}, \quad (3)$$

where $h_{t,i}(\widetilde{X}_t; \theta_{t-1}^P)$ captures the $i$-th token level internal states representation at the point when the $i$-th token error occurs. Then, to quantify the token-level error of the Pilot model, we compute the discrepancy between the predicted distribution $\hat{p}_{t,i}$ and the ground-truth distribution $p_{t,i}$ for each token, with the error defined as:

$$\ell_t(p_t, \hat{p}_t) = \left\{ \ell_t(p_{t,i}, \hat{p}_{t,i}) \right\}_{i=1}^n, \quad \text{with } \ell_t(p_{t,i}, \hat{p}_{t,i}) = p_{t,i} - \hat{p}_{t,i}. \quad (4)$$

Consistent with standard LLM fine-tuning procedures, where the loss $\mathcal{L}_t^P$ is used to compute gradients and update the Pilot model's parameters across $T$ rounds, we simultaneously collect key intermediate signals described above into the Mistake Log throughout this process. Formally, we define the Mistake Log as:

$$M_T = \left\{ \left( \widetilde{X}_t, \ h_t(\widetilde{X}_t; \theta_{t-1}^P), \ \ell_t(p_t, \hat{p}_t) \right) \right\}_{t=1}^T. \quad (5)$$

The Mistake Log systematically records contextual inputs, internal representations, and token-level prediction errors of the Pilot model throughout its entire fine-tuning trajectory. We next investigate how to leverage the Mistake Log during fine-tuning to enhance the Pilot model's final inference performance.

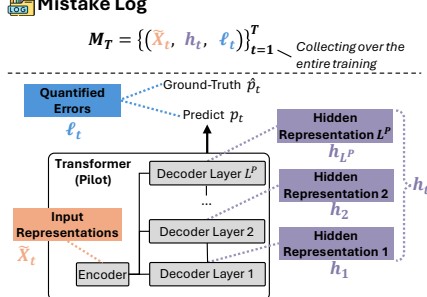

Figure 1: Illustration of the **Mistake Log**. We use the encoder-decoder architecture as an example here.

**Motivation for Transformer Copilot.** Recall that the goal of SFT is to optimize $\theta^P$ by minimizing the expected loss $\mathbb{E}_{(X_t, Y_t) \sim \mathcal{D}} \left[ \mathcal{L}_t^P \right]$. While this process adjusts model parameters using gradient descent, it treats each error as a transient signal, consumed and discarded immediately after the parameter update. As a result, the final model parameters $\theta_T^P$ might not retain an explicit memory of where, how, or why errors occurred during the training trajectory. This oversight leaves valuable training-time information, which we captured in the Mistake Log, untapped at inference time. To address this, we propose a new *Copilot* model to learn from the Mistake Log. Rather than altering the Pilot's optimization path, the Copilot operates as an auxiliary module that internalizes the distribution of past mistakes and corrects the Pilot's output at inference time. This design enables the Copilot to assist the Pilot model by reflecting on prior missteps and adaptively revising the predictions.

## 3 Transformer Copilot

We introduce our proposed framework, Transformer Copilot, which is designed for both encoder-decoder and decoder-only Transformer architectures. In the following sections, we will elaborate on the Copilot model design, the training paradigm, and the inference paradigm, respectively.

### 3.1 The Copilot Model Design

The Copilot model is initialized from the decoder module of the corresponding Pilot model, but with several new architectural modifications. Consistent with the Pilot model $f^P$, we denote the Copilot model as $f^C$, parameterized by $\theta^C$. The Copilot model is also auto-regressive, generating outputs

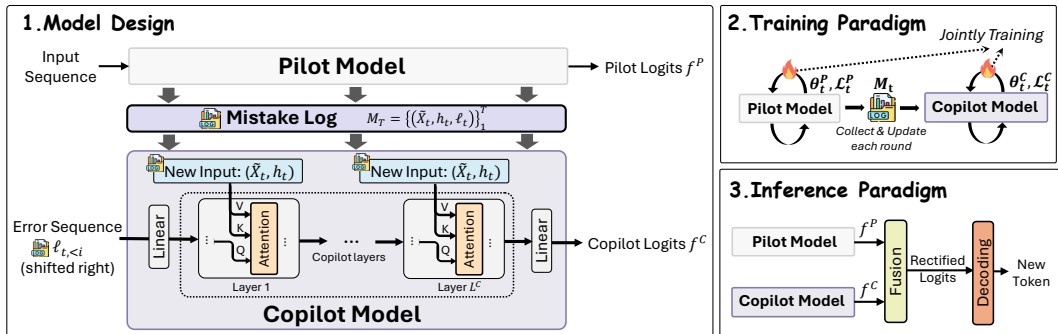

Figure 2: Transformer Copilot Framework. The overall framework comprises three key components: **(1) Copilot Model Design**, **(2) Training Paradigm**, and **(3) Inference Paradigm**.

over the vocabulary $V$. However, the objective of the Copilot model is to learn from the Mistake Log $M_T$ and output rectified logits that correct the predictions made by the Pilot model. Below, we specify the Copilot model design for the encoder-decoder and decoder-only Pilot model separately.

**Encoder-Decoder Copilot.** As shown in Figure 2.1, the Copilot model receives its inputs from the Mistake Log, $M_T = \{(\widetilde{X}_t, h_t(\widetilde{X}_t; \theta^P_{t-1}), \ell_t(p_t, \hat{p}_t))\}^T_{t=1}$. Specifically, the Copilot is conditioned on the sequence of token-level errors made by the Pilot model, as recorded in $M_T$, i.e. $\ell_{t,<i} = (p_{t,1} - \hat{p}_{t,1}, \ldots, p_{t,i-1} - \hat{p}_{t,i-1})$. These discrepancy sequences are provided as labels during training from $M_T$ and are auto-regressively generated during inference. As positional information is inherently preserved through the Pilot's output, we apply a single linear layer to project the token-level errors from vocabulary space into the Copilot's hidden dimension. Next, to incorporate additional information from the Pilot's input and internal hidden representations ($\widetilde{X}_t$ and $h_t$ from $M_T$), we propose a modified *cross-attention mechanism* in each layer of the Copilot, defined as:

$$\text{New } Q = H^C_{l-1} \cdot W^Q, \text{ for } l = 1, ..., L^C,$$
$$\text{New } K = \text{Concat}\big(\widetilde{X}_t, \text{ Pool}_{L^P}\big(h_t(\widetilde{X}_t; \theta^P_{t-1})\big)\big) \cdot W^K, \quad (6)$$
$$\text{New } V = \text{Concat}\big(\widetilde{X}_t, \text{ Pool}_{L^P}\big(h_t(\widetilde{X}_t; \theta^P_{t-1})\big)\big) \cdot W^V,$$

where $\text{Pool}_{L^P}(\cdot)$ denotes the mean pooling across $L^P$ layers of the Pilot and $\text{Concat}(\cdot)$ indicates concatenation along the sequence dimension to ensure input dimensional compatibility and computational efficiency; $H^C_{l-1}$ is the Copilot model's hidden state from the previous layer (or input projection layer at $l = 1$); and $W^Q, W^K, W^V$ are learnable attention weights. We then apply the standard scaled dot-product attention using the new $Q$, $K$, and $V$. This modified attention allows the Copilot to jointly attend to both the external input context and the internal processing dynamics of the Pilot. Note that all components retrieved from the Mistake Log can be directly accessed during the forward pass of the Pilot model, without incurring additional computational overhead. After the final layer $L^C$, we add a linear projection layer in the Copilot model to map the residual hidden representation into the vocabulary space, producing rectified logits as the output.

**Decoder-only Copilot.** We slightly adapt the Copilot model to accommodate the corresponding decoder-only Transformer [76, 1], while keeping the majority of the model input and design above unchanged. Specifically, we modify the *self-attention mechanism* to incorporate the information from the Mistake Log: In the odd-numbered layers of $L^C$, we retain the standard self-attention to allow the Copilot model to capture intra-sequence dependencies; In the even-numbered layers, we replace self-attention with the modified cross-attention mechanism defined in Eq. 6, enabling the Copilot to attend to the Pilot's input and internal state representations stored in $M_T$. This alternating structure is consistent with the encoder-decoder Copilot to capture its own error-correction dynamics and attend to informative signals from the Pilot's behavior. We also explore several alternative designs and empirically validate the effectiveness of our proposed design against these variants in Appendix F.4.

**Learning Objective.** Give the sequence pair $(X_t, Y_t)$, at $t$-th round, the objective of training the Copilot model $f^C$ at $i$-th token is defined as:

$$\mathcal{L}^C_t = \sqrt{\sum^n_{i=1} \|f^C_{t,i} - \ell_t(p_{t,i}, \hat{p}_{t,i})\|^2}, \quad \text{with } f^C_{t,i} = f^C(\widetilde{X}_t, h_{t,<i}, \ell_{t,<i}; \theta^C_{t-1}), \quad (7)$$

---
**Algorithm 1:** Transformer Copilot (Training Paradigm)
---

**Input:** Pilot model $f^P(\cdot; \theta^P)$, Copilot model $f^C(\cdot; \theta^C)$; Learning rates $\eta_P, \eta_C$; $T, n$

1   Initialize $\theta_0^P, \theta_0^C, M_0 \leftarrow \emptyset$
2   **for** $t = 1, 2, \ldots, T$ **do**
3     Draw $(X_t, Y_t) \sim \mathcal{D}$
4     $\triangledown$ Pilot - token-level forward pass
5     **for** $i = 1, \ldots, n$ **do**
6       Compute $\hat{p}_{t,i}$ via Eq.1
7     **end**
8     $\triangledown$ Collect Mistake Log (§2.2)
9     $M_t \leftarrow M_{t-1} \cup (\widetilde{X}_t, h_t(\widetilde{X}_t; \theta_{t-1}^P), \ell_t(p_t, \hat{p}_t))$
10     Compute $\mathcal{L}_t^P$ via Eq.2;
11     Update $\theta_t^P \leftarrow \theta_{t-1}^P - \eta_P \nabla_{\theta_{t-1}^P} \mathcal{L}_t^P$
12     /* For brevity, we reuse notation $t$ */
13     Draw $(\widetilde{X}_t, h_t(\widetilde{X}_t; \theta_{t-1}^P), \ell_t(p_t, \hat{p}_t)) \sim M_t$
14     $\triangledown$ Copilot - learn from the Mistake Log (§3.1)
15     **for** $i = 1, \ldots, n$ **do**
16       $f_{t,i}^C \leftarrow f^C(\widetilde{X}_t, h_{t,<i}, \ell_{t,<i}; \theta_{t-1}^C)$
17     **end**
18     Compute $\mathcal{L}_t^C$ via Eq.7;
19     Update $\theta_t^C \leftarrow \theta_{t-1}^C - \eta_C \nabla_{\theta_{t-1}^C} \mathcal{L}_t^C$
20   **end**
21   **return** $\theta_T^P, \theta_T^C$

---
**Algorithm 2:** Inference Paradigm
---

**Input:** $\theta_T^P, \theta_T^C$; Tuning parameter $\lambda$

1   Draw new $X_t \sim \mathcal{D}_\mathcal{X}, t > T$
2   **for** $i = 1, \ldots, n$ **do**
3     $\hat{p}_{t,i} \leftarrow \text{softmax}(f^P(X_t, \hat{y}_{t,<i}; \theta_T^P))$
4     Observe $\widetilde{X}_t, h_{t,<i}$ from $f^P$
5     $f_{t,i}^C \leftarrow f^C(\widetilde{X}_t, h_{t,<i}, f_{t,<i}^C; \theta_T^C)$
6     $\tilde{p}_{t,i} \leftarrow \hat{p}_{t,i} + \lambda f_{t,i}^C$ (via Eq.8)
7     $\hat{y}_{t,i} \leftarrow \text{Decoding}(\tilde{p}_{t,i})$
8   **end**
9   **return** $(\hat{y}_{t,1}, \ldots, \hat{y}_{t,n})$

where $f_{t,i}^C$ is the Copilot model's prediction, $\ell_t(p_{t,i}, \hat{p}_{t,i}) = p_{t,i} - \hat{p}_{t,i}$ is the corresponding label for the Copilot model, and $h_{t,<i}$ is the collection of Pilot's hidden states for the preceding tokens. We adopt the RMSE loss to prevent the distribution error from being further diminished by the square operation, avoiding the over-smoothing effect that squaring may introduce in the gradient signal during backpropagation. Next, we show how to jointly train the Pilot model $f^P$ and the Copilot model $f^C$ during fine-tuning, and collaborate on the generation during inference.

### 3.2 Training Paradigm.

Algorithm 1 outlines the process for jointly training the Pilot and Copilot model. In training round $t \in [T]$, one sequence pair $(X_t, Y_t)$ is drawn from the data distribution $\mathcal{D}$. For each token $i \in [n]$, we first compute the Pilot model's output distribution $\hat{p}_{t,i}$ (Line 5-7). We then retrieve information directly from the forward pass of the Pilot model and update the Mistake log $M_t$ by recording $\widetilde{X}_t$, $h_t$, and $\ell_t$ for each token (Line 9). Meanwhile, we compute the Pilot model's cross-entropy loss $\mathcal{L}_t^P$ and update its parameters (Lines 10-11). Next, we prepare the input for training the Copilot model. Given all collected previous training rounds' information, we draw a sample $(\widetilde{X}_t, h_t, \ell_t)$ from the updated mistake log $M_t$ (Line 13). We obtain the Copilot model's output $f_{t,i}^C$ for each token $i \in [n]$ (Line 15-17). Finally, we compute the Copilot model's RMSE loss $\mathcal{L}_t^C$ and update its parameters (Line 18-19). After $T$ rounds of iterative training, we obtain the final $\theta_T^P$ and $\theta_T^C$ for the Pilot and Copilot model, respectively. Note that this fine-tuning process can be readily extended to mini-batch stochastic gradient descent for scalability.

### 3.3 Inference Paradigm

After learning from the Mistake Log, the Copilot model is deployed alongside the Pilot model to enhance inference-time generation. To avoid abuse of notation, we reuse the same symbols as in training. Given a new input sequence $X_t \sim \mathcal{D}_\mathcal{X}, t > T$, where $X_t$ is not part of the training data, $t$ indexes the inference-time inputs and does not correspond to training rounds. As the objective of the Copilot model is to predict the token-level probability discrepancy $p_{t,i} - \hat{p}_{t,i}$, we directly use the Copilot model's output to rectify the Pilot model's prediction $\hat{p}_{t,i}$ towards the ground-truth $p_{t,i}$. Formally, the rectified predicted distribution is given by:

$$\tilde{p}_{t,i} = \hat{p}_{t,i} + \lambda f_{t,i}^C, \tag{8}$$

where $\lambda$ (typically set to 1) is a tunable hyperparameter controlling correction strength. Introducing $\lambda$ at inference allows for more flexible modulation, and as we later show in Section 4, with a proper $\lambda$, the rectified $\tilde{p}_{t,i}$ theoretically provides a closer approximation to the target distribution $p_{t,i}$. Algorithm 2 outlines the overall inference paradigm. Given $X_t$, the Pilot model outputs a predicted distribution $\hat{p}_{t,i}$ at each token generation step $i \in [n]$ (Line 3). Subsequently, the Copilot model auto-regressively computes its output $f_{t,i}^C$ (Line 5). Finally, the rectified $\tilde{p}_{t,i}$ is obtained via Eq.8 and used to generate the next token via a decoding function (Lines 6-7). The inference process is adaptive and can optionally terminate upon generation of the [EOS] (end-of-sequence) token.

## 4  Analyses - Why Learn from the Mistake Log?

To elucidate the roles of the Mistake Log and Copilot model in enhancing the Pilot model's inference-time performance, we present both theoretical and empirical analyses in this section.

**Theoretical Guarantee.** Recall that the Copilot model $f^C$ is designed to analyze the Pilot model's internal cognitive states $\widetilde{X}_t, h_t$ via the collected Mistake Log $M_T$, and learns to predict errors measured by the token-level discrepancies $\ell_t(p_{t,i}, \hat{p}_{t,i})$. During inference, we use the rectified prediction as $\tilde{p}_{t,i} = \hat{p}_{t,i} + \lambda f_{t,i}^C$. In the following analysis, we show that, under mild assumptions, the adjusted prediction $\tilde{p}_{t,i}$ yields improved inference performance over the original estimate $\hat{p}_{t,i}$.

Let $A^P, A^C$ denote the distributions over the function classes of $\theta^P, \theta^C$, induced by the randomness in the fine-tuning process. Let $[k]$ denote the $k$-th dimension of a vector in $\mathbb{R}^{|V|}$. Then, we define the expected error and variance of the Pilot and Copilot model at the $k$-th output dimension as:

$$\epsilon_P^2 = \mathbb{E}_{(X_t, Y_t)\sim\mathcal{D}} \left[ (p_{t,i}[k] - \mathbb{E}_{\theta^P\sim A^P}[\hat{p}_{t,i}[k] \mid \hat{y}_{t,<i}])^2 \right],$$

$$\sigma_P^2 = \mathbb{E}_{(X_t, Y_t)\sim\mathcal{D}} \left[ \mathrm{Var}_{\theta^P\sim A^P}[\hat{p}_{t,i}[k] \mid \hat{y}_{t,<i}] \right],$$

$$\epsilon_C^2 = \mathbb{E}_{\substack{\theta^P\sim A^P \\ (X_t, Y_t)\sim\mathcal{D}}} \left[ \left( p_{t,i}[k] - \hat{p}_{t,i}[k] - \mathbb{E}_{\theta^C\sim A^C}[f_{t,i}^C[k] \mid f_{t,<i}^C] \right)^2 \mid \hat{y}_{t,<i} \right],$$

$$\sigma_C^2 = \mathbb{E}_{\substack{\theta^P\sim A^P \\ (X_t, Y_t)\sim\mathcal{D}}} \left[ \mathrm{Var}_{\theta^C\sim A^C}[f_{t,i}^C[k] \mid f_{t,<i}^C] \mid \hat{y}_{t,<i} \right].$$

**Theorem 4.1.** *For any $k \in [|V|]$, suppose that $\epsilon_P^2 > 0$ and $\epsilon_C < \sqrt{\epsilon_P^2 + \sigma_P^2}$. Then there exists $\lambda_0 > 0$ such that for any $0 < \lambda < \lambda_0$, the rectified prediction $\tilde{p}_{t,i} = \hat{p}_{t,i} + \lambda f_{t,i}^C$ yields a strictly closer approximation to the ground-truth distribution $p_{t,i}$ at dimension $k$. Specifically, at the $i$-th token prediction step for $X_t \sim \mathcal{D}_{\mathcal{X}}$, we have:*

$$\mathbb{E}_{\substack{\theta^P\sim A^P \\ \theta^C\sim A^C \\ (X_t, Y_t)\sim\mathcal{D}}} \left[ (p_{t,i}[k] - \tilde{p}_{t,i}[k])^2 \mid f_{t,<i}^C, \hat{y}_{t,<i} \right] < \mathbb{E}_{\substack{\theta^P\sim A^P \\ (X_t, Y_t)\sim\mathcal{D}}} \left[ (p_{t,i}[k] - \hat{p}_{t,i}[k])^2 \mid \hat{y}_{t,<i} \right].$$

*Remark* 4.2. The assumption $\epsilon_C < \sqrt{\epsilon_P^2 + \sigma_P^2}$ in Theorem 4.1 allows the Copilot model $f^C(\cdot; \theta^C)$ to have a larger bias than the bias $\epsilon_P$ of the Pilot model $f^P(\cdot; \theta^P)$, i.e., $\epsilon_P^2 < \epsilon_C^2 < \epsilon_P^2 + \sigma_P^2$.

Theorem 4.1 suggests that the rectified prediction $\tilde{p}_{t,i}$ after incorporating the Copilot model achieves strictly lower expected error at $k$-dimension under mild assumptions and a proper $\lambda$, indicating the Copilot helps improve the inference performance of the Pilot. The full proof is provided in Appendix B. In addition, Remark 4.2 implies that the Copilot model can improve inference performance without needing to match the Pilot's accuracy in isolation. This insight motivates us to apply a relatively smaller size of a Copilot to complement the Pilot in our empirical implementation.

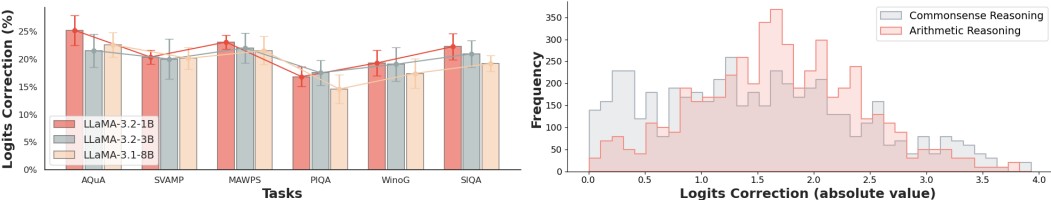

Figure 3: **Logits Correction by Copilot.** We visualize the logits correction introduced by a 1B Copilot model (computed as |*Fused logits* − *Pilot logits*|) to highlight the **shift** by the Copilot's rectification. **Left:** Percentage of logits correction over original Pilot's output logits range for three LLaMA-3 Pilot models. **Right:** Distribution of logits correction magnitudes across reasoning types.

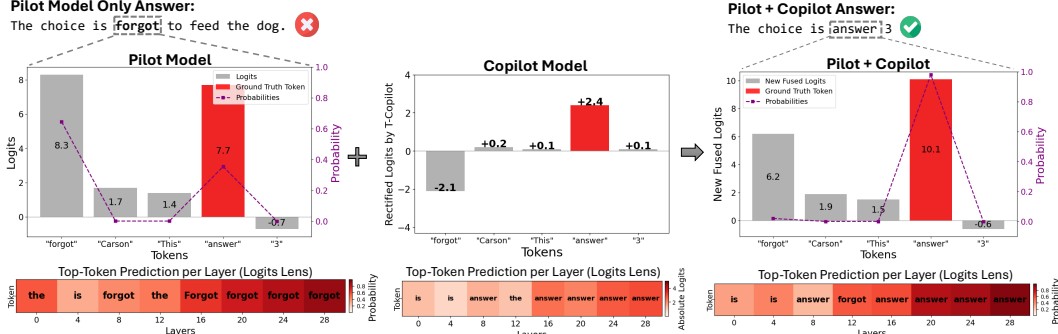

Figure 4: **Example of Copilot's Token-level Rectification on SIQA.** The token-level formatting error ('forgot') originates during the Pilot's mid-way generation and is corrected ('answer') by incorporating the Copilot.

**Empirical Analysis.** Complementing our theoretical analysis, we empirically examine the rectification effectiveness of the Copilot model during inference. We leave the setups in Appendix C. Figure 3 illustrates the average logits correction induced by the 1B Copilot model across different Pilot models and reasoning categories. Given that the typical logits range is approximately $[-10, 10]$, the observed shifts on the logits distribution indicate a clear and consistent adjustment on the final predictions by the Copilot model.

We further verify that this Copilot's adjustment indeed steers the token prediction toward the correct direction: We analyze representative error patterns frequently observed in the Pilot model's output, particularly factual and formatting mistakes. Figure 4 shows a detailed example of token-level logits rectification on Pilot model LLaMA-3.2-3B by the 1B Copilot, visualized using the layer-wise Logits Lens [8]. At mid-inference, the Pilot does not follow the correct answer format and makes mistakes (the correct token 'answer' has a high but suboptimal logit). The Copilot rectifies the prediction by decreasing the logit of the incorrect token 'forgot' and amplifying that of the correct token, thereby correcting the token prediction error. We leave analyses on other error patterns in Appendix C.

## 5    Empirical Evaluations

**Tasks and Datasets.** To comprehensively evaluate T-Copilot, we utilize a broad suite of reasoning and generation tasks: (i) Commonsense reasoning: PIQA [10], HellaSwag [93], WinoGrande [68], BoolQ [18], SIQA [70], and OpenbookQA (OBQA) [55]. (ii) Arithmetic reasoning: AQuA [48], GSM8K [19], MAWPS [42], and SVAMP [60]. and (iii) Downstream Recommendation: Beauty [30] and LastFM [67]. Detailed dataset descriptions are provided in Appendix D.

**Implementation Details.** For T-Copilot, we construct the Copilot model using the same type of decoder architecture as the Pilot model to ensure consistency. We use the AdamW optimizer and Cosine learning rate scheduler for both Pilot and Copilot models. We modify the `generate` in HuggingFace Transformers [22] to perform token-level logits fusion and rectified next-token generation during inference. All experiments are conducted on NVIDIA A100 GPUs. We leave all hyperparameter setups and training/inference details in Appendix E.1.

**Models and Baselines.** We incorporate T-Copilot with varying backbone Pilot models. For encoder-decoder Pilots, we utilize T5 [65] and FLAN-T5 [17] across small/base/large variants. For decoder-only Pilots, we employ multiple models from LLaMA-3 [21] and Qwen2.5 [90] families. We denote **T-Copilot-small/base/0.5B/1B/3B** as the Copilot model on different scales. Detailed model configuration and implementation details are provided in Appendix E.2. We compare against three baseline types: (i) Pilot-only models as described above. (ii) Frontier LLMs with comparable and larger parameters, including LLaMA-3.1-8B [21], Gemma-2-9B [75], and Qwen2.5-14B. (iii) Layer/Adapter expansion methods, including MoE models [71] (Mistral-7B, Ministral-8B), LLaMA/Mistral-Pro-8B [84], Mergekit-9B [26], and TIES [89]. Detailed baseline descriptions are provided in Appendix E.3.

### 5.1    Incorporating T-Copilot into Pilot Models Yields Better Performance

**Effectiveness of Copilot in Enhancing Pilot.** Table 1 presents the performance gains of incorporating T-Copilot into the Pilot models across different model scales and types. T-Copilot consistently

Table 1: Experiment results (%) of incorporating T-Copilot on encoder-decoder/decoder-only backbone models. Results are averaged over 3 independent runs. We report the relative improvement on the backbone Pilot models. T-Copilot boosts existing LLMs on ten reasoning tasks by 2.0%–34.5%.

| Type | Model | Commonsense Reasoning (Acc. ↑) | | | | | | | Impr. | Arithmetic Reasoning (Acc. ↑) | | | | | Impr. |
|------|-------|------|--------|--------|------|------|------|------|-------|------|-------|-------|-------|------|-------|
| | | PIQA | WinoG. | HellaS. | BoolQ | SIQA | OBQA | Avg. | | AQuA | GSM8K | MAWPS | SVAMP | Avg. | |
| T5 | FLAN-T5-small | 60.3 | 52.1 | 31.6 | 57.9 | 47.8 | 29.2 | 46.5 | | 19.6 | 5.6 | 14.7 | 5.3 | 11.3 | |
| | **+ T-Copilot-small** | 63.1 | 54.4 | 34.9 | 61.7 | 52.7 | 32.9 | 50.0 | ↑**7.5%** | 24.8 | 7.4 | 20.6 | 8.0 | 15.2 | ↑**34.5%** |
| | FLAN-T5-base | 65.4 | 54.6 | 36.8 | 61.1 | 48.6 | 29.6 | 49.4 | | 22.8 | 7.2 | 27.1 | 6.3 | 15.9 | |
| | **+ T-Copilot-base** | 67.3 | 56.2 | 39.7 | 62.5 | 54.3 | 34.7 | 52.5 | ↑**6.3%** | 24.4 | 9.3 | 32.4 | 10.3 | 19.1 | ↑**20.1%** |
| | FLAN-T5-large | 70.5 | 60.4 | 49.5 | 62.2 | 58.1 | 31.7 | 55.4 | | 23.2 | 9.9 | 36.7 | 9.7 | 19.9 | |
| | **+ T-Copilot-small** | 72.2 | 61.9 | 51.3 | 63.2 | 59.8 | 32.6 | 56.8 | ↑**2.5%** | 24.7 | 11.3 | 37.2 | 11.6 | 21.2 | ↑**6.5%** |
| | **+ T-Copilot-base** | 72.8 | 63.6 | 52.3 | 63.7 | 60.8 | 34.2 | 57.9 | ↑**4.5%** | 25.1 | 11.6 | 39.8 | 13.8 | 22.6 | ↑**13.6%** |
| LLaMA | LLaMA-3.2-1B | 77.5 | 71.1 | 61.8 | 63.9 | 71.9 | 66.8 | 68.8 | | 25.6 | 27.3 | 77.1 | 47.3 | 44.3 | |
| | **+ T-Copilot-1B** | 80.2 | 73.7 | 63.3 | 65.5 | 74.9 | 68.9 | 71.1 | ↑**3.3%** | 28.3 | 32.2 | 81.5 | 51.6 | 48.4 | ↑**9.3%** |
| | LLaMA-3.2-3B | 83.3 | 79.6 | 89.4 | 69.1 | 77.4 | 75.6 | 79.1 | | 33.1 | 55.3 | 86.1 | 64.2 | 59.7 | |
| | **+ T-Copilot-1B** | 84.1 | 82.6 | 91.1 | 70.3 | 78.6 | 77.2 | 80.7 | ↑**2.0%** | 36.6 | 58.2 | 89.1 | 68.7 | 63.2 | ↑**5.9%** |
| | **+ T-Copilot-3B** | 85.6 | 83.7 | 91.3 | 72.8 | 79.2 | 81.3 | 82.3 | ↑**4.0%** | 40.1 | 63.1 | 91.2 | 71.4 | 66.5 | ↑**11.4%** |
| | LLaMA-3.1-8B | 85.4 | 84.3 | 90.9 | 69.6 | 79.9 | 82.6 | 82.1 | | 37.3 | 63.5 | 89.1 | 73.6 | 65.9 | |
| | **+ T-Copilot-1B** | 86.2 | 86.8 | 93.5 | 71.8 | 82.7 | 83.2 | 84.0 | ↑**2.3%** | 38.9 | 66.1 | 90.8 | 75.4 | 67.8 | ↑**2.9%** |
| Qwen | Qwen2.5-3B | 83.6 | 77.5 | 89.8 | 63.4 | 77.6 | 84.6 | 79.4 | | 55.9 | 71.4 | 89.6 | 81.5 | 74.6 | |
| | **+ T-Copilot-0.5B** | 85.4 | 79.1 | 91.3 | 66.8 | 78.1 | 86.0 | 81.1 | ↑**2.1%** | 57.3 | 74.2 | 91.8 | 82.8 | 76.5 | ↑**2.5%** |
| | **+ T-Copilot-3B** | 87.8 | 81.7 | 94.0 | 68.7 | 79.9 | 89.4 | 83.6 | ↑**5.3%** | 59.4 | 76.8 | 92.6 | 83.5 | 78.1 | ↑**4.7%** |
| | Qwen2.5-7B | 87.2 | 82.1 | 91.4 | 71.2 | 79.3 | 89.1 | 83.4 | | 61.0 | 75.3 | 91.2 | 84.8 | 78.1 | |
| | **+ T-Copilot-0.5B** | 89.3 | 85.3 | 93.5 | 73.6 | 80.0 | 92.1 | 85.6 | ↑**2.6%** | 61.4 | 78.2 | 93.0 | 86.5 | 79.8 | ↑**2.2%** |
| | **+ T-Copilot-3B** | 92.5 | 87.2 | 95.3 | 74.8 | 84.3 | 94.9 | 88.2 | ↑**5.8%** | 64.2 | 79.7 | 94.8 | 88.1 | 81.7 | ↑**4.6%** |

Table 2: Performance comparison (%) with baselines **under matched parameter scales**. Results are averaged over 3 runs. Adding T-Copilot consistently surpasses baselines of equal or even larger size.

| Model | Params | Commonsense Reasoning (Acc. ↑) | | | | | | | Arithmetic Reasoning (Acc. ↑) | | | | |
|-------|--------|------|--------|--------|------|------|------|------|------|-------|-------|-------|------|
| | | PIQA | WinoG. | HellaS. | BoolQ | SIQA | OBQA | Avg. | AQuA | GSM8K | MAWPS | SVAMP | Avg. |
| LLaMA-3.1-8B | 8B | 85.4 | 84.3 | 90.9 | 69.6 | 79.9 | 82.6 | 82.1 | 37.3 | 63.5 | 89.1 | 73.6 | 65.9 |
| LLaMA-3.2-3B + T-Copilot-3B | 6B (-2B) | 85.6 | 83.7 | 91.3 | 72.8 | 79.2 | 81.3 | **82.3** | 40.1 | 63.1 | 91.2 | 71.4 | **66.5** |
| Qwen2.5-7B | 7B | 87.2 | 82.1 | 91.4 | 71.2 | 79.3 | 89.1 | 83.4 | 61.0 | 75.3 | 91.2 | 84.8 | **78.1** |
| Qwen2.5-3B + T-Copilot-3B | 6B (-1B) | 87.8 | 81.7 | 94.0 | 68.7 | 79.9 | 89.4 | **83.6** | 59.4 | 76.8 | 92.6 | 83.5 | **78.1** |
| Qwen2.5-14B | 14B | 91.8 | 85.6 | 94.3 | 75.2 | 84.5 | 93.1 | 87.4 | 63.5 | 79.5 | 92.4 | 87.9 | 80.8 |
| Qwen2.5-7B + T-Copilot-3B | 10B (-4B) | 92.5 | 87.2 | 95.3 | 74.8 | 84.3 | 94.9 | **88.2** | 64.2 | 79.7 | 94.8 | 88.1 | **81.7** |
| ***Comparison with Layer/Adapter Expansion Baselines*** | | | | | | | | | | | | | |
| Mistral-Pro-8B | 8B | 83.1 | 81.9 | 86.1 | 70.8 | 76.1 | 80.6 | 79.8 | 35.5 | 54.4 | 88.2 | 68.5 | 61.7 |
| LLaMA-Pro-8B | 8B | 88.4 | 81.4 | 86.9 | 73.9 | 76.1 | 77.8 | 80.8 | 38.2 | 57.2 | 92.5 | 63.5 | 62.9 |
| Ministral-8B | 8B | 85.7 | 84.1 | 91.3 | 70.3 | 77.5 | 81.3 | 81.7 | 37.4 | 62.9 | 90.2 | 73.2 | 65.9 |
| LLaMA-3.2-3B + T-Copilot-3B | 6B (-2B) | 85.6 | 83.7 | 91.3 | 72.8 | 79.2 | 81.3 | **82.3** | 40.1 | 63.1 | 91.2 | 71.4 | **66.5** |
| MergeKit-9B | 9B | 86.1 | 84.7 | 91.1 | 71.1 | 79.3 | 80.2 | 82.1 | 37.0 | 65.2 | 90.3 | 75.2 | 66.9 |
| LLaMA-3.1-8B + T-Copilot-1B | 9B | 86.2 | 86.8 | 93.5 | 71.8 | 82.7 | 83.2 | **84.0** | 38.9 | 66.1 | 90.8 | 75.4 | **67.8** |

improves performance across all T5, LLaMA, and Qwen models on 10 commonsense and arithmetic reasoning tasks. In particular, a lightweight Copilot (e.g., T-Copilot-small) can deliver meaningful improvements (6.5% on arithmetic) when paired with a much larger Pilot model (e.g., FLAN-T5-large). Moreover, scaling up the Copilot model leads to additional improvement, underscoring its effectiveness in rectifying the Pilot model's predictions during inference.

**Comparison with Size-Matched Baselines.** As shown in Table 2, we first compare our method against stronger models with larger parameters under the same model backbones. While LLaMA-3.2-3B initially lags significantly behind LLaMA-3.1-8B, incorporating T-Copilot-3B enables the model to outperform LLaMA-3.1-8B, despite using 2B fewer total parameters. Similarly, for the Qwen2.5 series, incorporating T-Copilot-3B enables the smaller Qwen2.5-7B to surpass Qwen2.5-14B with 4B fewer parameters. To provide a broader perspective, we also compare with strong baselines from different methods and model types. For instance, although LLaMA-3.2-3B originally trails behind models like Ministral-8B and LLaMA-Pro-8B, incorporating T-Copilot-3B enables it to outperform the strongest baseline under the 8B scale, Ministral-8B, while maintaining a 2B parameter advantage. Due to page limits, full comparison results are provided in Appendix F.1.

**Downstream Tasks.** Additional evaluation of T-Copilot and baseline comparisons on downstream recommendation tasks is provided in Appendix F.2.

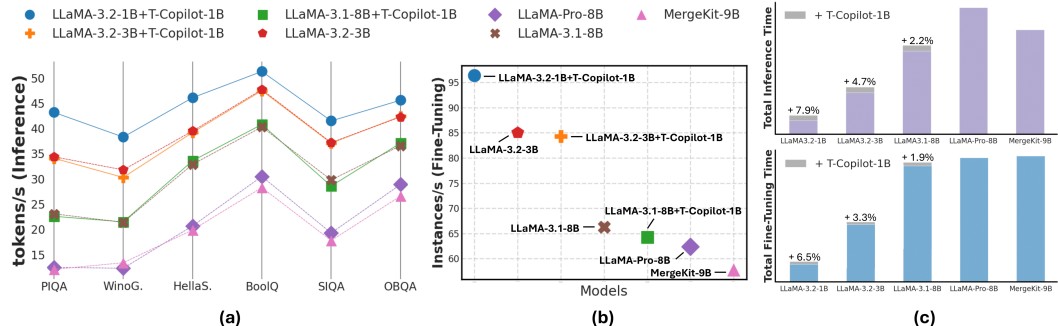

Figure 5: Efficiency Analysis on T-Copilot during fine-tuning and inference. **(a)** Inference model throughput. **(b)** Fine-tuning running speeds. **(c)** Overall training and inference time overhead.

Table 3: **T-Copilot Transferability Results.** We report the performance of T-Copilot paired with new Pilot models across four reasoning tasks. The results demonstrate that the Copilot model remains effective for the new Pilot models without being jointly trained.

| T-Copilot-1B | HellaSwag | BoolQ | GSM8K | SVAMP | **Overall Impr.** |
|---|---|---|---|---|---|
| *with* new LLaMA-3.2-1B | 63.1 | 65.2 | 32.2 | 51.4 | ↑**6.1%** |
| *with* new LLaMA-3.3-3B | 91.4 | 70.2 | 58.8 | 68.5 | ↑**4.2%** |
| *with* new LLaMA-3.1-8B | 93.1 | 71.7 | 66.0 | 75.8 | ↑**2.4%** |

## 5.2 Efficiency, Transferability, and Scalability

**Efficiency.** To thoroughly evaluate T-Copilot's running efficiency, we compare against Pilot and baseline models with the same LLaMA-3 backbone architecture under similar parameter scales. As shown in Figure 5, T-Copilot maintains comparable inference throughput (Figure 5 (a)) and training speed (Figure 5 (b)) to its corresponding Pilot models, while incurring only a 4% marginal average increase in time overhead (Figure 5 (c)). In contrast, other baselines such as LLaMA-Pro-8B and MergeKit-9B suffer from significantly higher latency and computational costs relative to their base model LLaMA-3.1-8B. We provide a more detailed inference latency report in Appendix F.3 (Table 15) and discuss the architectural advantage of our model design in Appendix A.1.

**Transferability.** In the T-Copilot learning framework, the Copilot model is fine-tuned alongside but separately from the Pilot model. Since the same type of models generally have similar learning trajectories under identical training settings, we further investigate if the Copilot model can leverage the mistake log of one Pilot model and still be effective on another Pilot model of the similar type.

We conduct controlled experiments on LLaMA-3 series models in which we directly apply a finetuned 1B Copilot model to new Pilot models during inference. The new Pilot model shares the same architecture as the original one but is trained independently. Note that the Copilot model does not "see" or "learn" any information from the new Pilot model, as they are not jointly trained during finetuning. In Table 3, transferring the Copilot model leads to a slight ±0.2% performance difference compared to applying the Copilot to the initial Pilot models (jointly training together). We hypothesize that the minor discrepancy is due to the hardware inference differences between the original and new Pilot models. Nonetheless, the transferred Copilot model still delivers substantial performance gains for the new Pilot and consistently outperforms competing baselines. These results demonstrate that T-Copilot's error-correction capabilities are not tightly coupled to a specific Pilot model and can be effectively transferred without additional rounds of fine-tuning.

**Scalability.** Figure 6 illustrates the relationship between accuracy and model parameter size for T-Copilot. Overall, incorporating the Copilot model consistently demonstrates improved performance as model size increases. We analyze the relationship between performance accuracy ($A$) and model parameter size ($N$) in billions. The derived equations for our method are as follows: for Flan-T5 backbones: $A \simeq 8.74 \cdot \log_{10}(N) + 40.17$; For LLaMA-3 backbones: $A \simeq 29.58 \cdot \log_{10}(N) + 50.20$, and for Qwen2.5 backbones: $A \simeq 12.40 \cdot \log_{10}(N) + 71.80$.

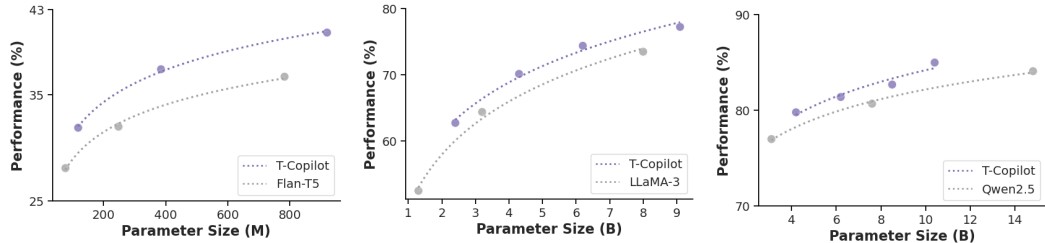

Figure 6: **Inference Scaling Laws** for T-Copilot. We evaluate the average accuracy of T-Copilot and backbone frontier LLMs across all reasoning tasks at varying model scales. The results are shown for three architectures: FLAN-T5 (left), LLaMA-3 (middle), and Qwen2.5 (right).

**Ablation Studies.** Detailed ablation studies on T-Copilot, including model design choices, input insertion patterns, and the effect of the hyperparameter $\lambda$, are presented in Appendix F.4.

## 6 Related Works

**LLMs Supervised Fine-tuning.** Supervised fine-tuning (SFT) serves as the standard post-training method for specializing pre-trained LLMs to downstream tasks [80, 69, 94]. It enables models to incorporate task-specific knowledge and improves their performance in domain-relevant settings [94, 59, 96]. While effective, SFT often suffers from misalignment between training-time objectives and inference-time behavior [56, 79], leading to suboptimal generalization. Recent work has explored parameter-efficient tuning methods [32, 46], alongside advanced adaptation strategies [74, 49, 85] that improve learning effectiveness and efficiency. These methods primarily focus on model capacity and optimization rather than leveraging learning dynamics. Building upon prior SFT methods, our approach is compatible with existing fine-tuning frameworks and further improves by incorporating model-internal signals into the fine-tuning process. By adaptively learning from mistake patterns observed during fine-tuning, T-Copilot enables error-aware prediction and helps reduce the gap between training and inference performance.

**Self-refinement in Language Models.** Recent research has explored various self-refinement techniques in LLMs to generate high-quality outputs. Models either iteratively prompt themselves with updated responses [52, 27, 72] or optimize their behavior using external human or synthetic feedback [58, 91, 54, 99]. Orthogonal to external supervision such as additional prompting, multi-stage feedback, or explicit reward optimization, our work focuses on capturing model-internal signals during fine-tuning to achieve token-level rectification, without modifying the training objective or data distribution. We leave the additional related work and discussions in the Appendix G.

## 7 Conclusion

In this paper, we introduce Transformer Copilot, a novel learning framework that enhances Transformer-based Pilot models by integrating an auxiliary Copilot model during fine-tuning. By capturing the Pilot model's learning signals in a Mistake Log during fine-tuning, the Copilot model learns to rectify the Pilot's logits at inference time, enabling error-aware predictions. We provide both theoretical and empirical evidence that our method improves the Pilot model's inference predictions. Experiments on 12 benchmarks demonstrate the effectiveness, efficiency, scalability, and transferability of Transformer Copilot. Discussions on limitations are provided in Appendix H.

## Acknowledgment

We would like to thank the anonymous reviewers and AC for their constructive feedback during the review process. We also thank Sirui Chen, Tianxin Wei, Zhining Liu, Xiao Lin and other members from UIUC iDEA-iSAIL Lab for their constructive feedback on this work. This work is supported by National Science Foundation under Award No. IIS-2117902, and Agriculture and Food Research Initiative (AFRI) grant no. 2020-67021-32799/project accession no.1024178 from the USDA National Institute of Food and Agriculture. The views and conclusions are those of the authors and should not be interpreted as representing the official policies of the funding agencies or the government.

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

# Appendix

## Table of Contents

# A  Additional Details on Transformer Copilot

## A.1  Architectural Advantages of T-Copilot.

In Section 3.1, we introduce the Copilot model inherited from the standard decoder module in a Transformer [77]. However, our model design exhibits several key advantages compared to the standard decoder module: Specifically, our Copilot model (i) eliminates the need for positional embeddings to preprocess input sequences, (ii) does not require a softmax layer to normalize high-dimensional logits distributions, and (iii) avoids waiting for the computation of key-value (KV) pairs from the previous layer. These architectural design choices distinguish our method from layer adaptation methods [26, 84] that modify internal Transformer layers, which inherently introduce additional computational overhead. As a result, our method minimizes the gap in efficiency between our framework and vanilla models.

## A.2  Decoder-only Copilot Details

The decoder-only copilot model $f^C$ inherits its structure from the pilot model and processes three inputs from the Mistake Log: the token-level discrepancy sequence $\ell_t$, the embedded input sequence $\widetilde{X}_t$, and the pilot model's hidden states $h_t$. Note that, different from the encoder-decoder Pilot model, $\widetilde{X}_t$ here is derived from the input sequence $X_t$ after the positional embedding layer.

In the Decoder-only Copilot model, as stated in Section 3.1, the alternating attention mechanisms effectively mirror the encoder-decoder structure, enabling the decoder-only Copilot to leverage information inside the Mistake Log corrected from the Pilot model. The loss function (RMSE) and target values $\ell_t(p_{t,i}, \hat{p}_{t,i})$ for the Decoder-only Copilot model remain identical to those used for the encoder-decoder Copilot version. The fine-tuning and inference paradigm are also the same as the encoder-decoder Copilot model, as stated in Algorithm 1 and 2.

## A.3  Buffer of the Mistake Log

As described in Section 3, we maintain a Mistake Log to record the Pilot model's internal learning signals, which serve as training data for the Copilot model. To store this information efficiently with minimal GPU and CPU memory overhead, we detach all relevant outputs from the Pilot's forward pass and store them in a CPU-resident buffer. By default, we use a fixed-size buffer that retains the most recent 128 training rounds. The buffer is updated at each training step, and all Copilot training samples are drawn exclusively from it. This design keeps the additional memory footprint lightweight, typically under 500MB of the CPU memory and less than 200MB of the GPU memory.

## A.4  Conceptual Rationale on T-Copilot's Learning Objective

We elaborate here on the motivation for adopting the RMSE loss in training T-Copilot. Recall that the objective of the Copilot is to predict the distribution discrepancy, defined as

$$\ell_t(p_{t,i}, \hat{p}_{t,i}) = p_{t,i} - \hat{p}_{t,i},$$

in our formulation. Note that $\ell_t(p_{t,i}, \hat{p}_{t,i})$ is not a valid probability distribution, since $\sum_i \ell_t(p_{t,i}, \hat{p}_{t,i}) \neq 1$. Therefore, it is natural to formulate this task as a **regression problem**, for which the **RMSE** loss is commonly adopted.

In contrast, common language modeling objectives such as Cross-Entropy (CE) and KL Divergence are designed for **distribution fitting** and are **not directly applicable** in this setting, unless we manipulate $\ell_t(p_{t,i}, \hat{p}_{t,i})$ with a softmax function to resemble a valid distribution. In that case, however, the loss would no longer optimize the original discrepancy (i.e., $p_{t,i} - \hat{p}_{t,i}$), leading to potential information loss. In Appendix F.4, we further empirically validate the effectiveness of the RMSE loss for T-Copilot training.

# B  Proof of Theorem 4.1

Given the model parameters $\theta^P$ and $\theta^C$, we denote the Pilot model as $f^P(\cdot; \theta^P)$ and the Copilot model as $f^C(\cdot; \theta^C)$. Let $X_t \sim \mathcal{D}_{\mathcal{X}}$ represent the input sequence at inference step $t$, and $\widetilde{X}_t$ be the input representation of the $X_t$; $Y_t$ be the corresponding ground-truth answer for the input sequence $X_t$. For the $t$-th token prediction during inference, recall that:

$$p_{t,i} = \mathbb{P}\left(y_{t,i} \mid X_t, \hat{y}_{t,<i}\right),$$
$$\hat{p}_{t,i} = \text{softmax}(f^P(X_t, \hat{y}_{t,<i}; \theta^P)),$$
$$f_{t,i}^C = f^C(\widetilde{X}_t, h_{t,<i}, f_{2,t,<i}; \theta^C).$$

Let $A^P$, $A^C$ denote the distributions over the function classes of $\theta^P, \theta^C$, induced by the randomness in the fine-tuning process. Let $[k]$ denote the $k$-th dimension of a vector in $\mathbb{R}^{|V|}$. Then, we define the expected error and variance of the Pilot and Copilot model at the $k$-th output dimension as:

$$\epsilon_P^2 := \mathbb{E}_{(X_t,Y_t)\sim\mathcal{D}}\left[(p_{t,i}[k] - \mathbb{E}_{\theta^P\sim A^P}[\hat{p}_{t,i}[k] \mid \hat{y}_{t,<i}])^2\right] < \infty,$$

$$\sigma_P^2 := \mathbb{E}_{(X_t,Y_t)\sim\mathcal{D}}\left[\text{Var}_{\theta^P\sim A^P}[\hat{p}_{t,i}[k] \mid \hat{y}_{t,<i}]\right] < \infty,$$

$$\epsilon_C^2 := \mathbb{E}_{\substack{\theta^P\sim A^P \\ (X_t,Y_t)\sim\mathcal{D}}}\left[(p_{t,i}[k] - \hat{p}_{t,i}[k] - \mathbb{E}_{\theta^C\sim A^C}[f_{t,i}^C[k] \mid f_{t,<i}^C])^2 \mid \hat{y}_{t,<i}\right] < \infty,$$

$$\sigma_C^2 := \mathbb{E}_{\substack{\theta^P\sim A^P \\ (X_t,Y_t)\sim\mathcal{D}}}\left[\text{Var}_{\theta^C\sim A^C}[f_{t,i}^C[k] \mid f_{t,<i}^C] \mid \hat{y}_{t,<i}\right] < \infty$$

where we assume $f^P$ and $f^C$ have the bounded error and variance at $k$-th dimension. Here, $p_{t,i}[k]$ denotes the ground-truth probability assigned to the token at dimension $k \in [|V|]$ of the vocabulary, for the $i$-th token prediction step within input sequence $X_t$. Then, we have the following theorem, which corresponds to Theorem 4.1 in the main body.

**Theorem B.1** (Restate). *Given $A^P$, $A^C$, the Pilot model $f^P(\cdot; \theta^P)$, the Copilot model $f^C(\cdot; \theta^C)$, and a data distribution $\mathcal{D}$. For any $k \in [|V|]$, suppose the Pilot model is imperfect, i.e., $\epsilon_P^2 > 0$, and the Copilot model's error satisfies $\epsilon_C < \sqrt{\epsilon_P^2 + \sigma_P^2}$. Then there exists a constant $\lambda_0 > 0$ such that for any $0 < \lambda < \lambda_0$, the rectified prediction $\tilde{p}_{t,i} = \hat{p}_{t,i} + \lambda f_{t,i}^C$ yields a strictly closer approximation to the ground-truth distribution $p_{t,i}$ at dimension $k$. Specifically, at the $i$-th token prediction step for $X_t \sim \mathcal{D}_{\mathcal{X}}$, we have:*

$$\mathbb{E}_{\substack{\theta^P\sim A^P \\ \theta^C\sim A^C \\ (X_t,Y_t)\sim\mathcal{D}}}\left[(p_{t,i}[k] - \tilde{p}_{t,i}[k])^2 \,\Big|\, f_{t,<i}^C, \hat{y}_{t,<i}\right] < \mathbb{E}_{\substack{\theta^P\sim A^P \\ (X_t,Y_t)\sim\mathcal{D}}}\left[(p_{t,i}[k] - \hat{p}_{t,i}[k])^2 \mid \hat{y}_{t,<i}\right].$$

*Proof.* For brevity, we omit the explicit expectation condition over the Pilot model's previously generated tokens $\hat{y}_{t,<i}$, the Copilot model's preceding outputs $f_{t,<i}^C$, and the dimension index $[k]$ in the following proof.

Firstly, by the law of total expectation w.r.t. $(X_t, Y_t)$ and the bias-variance decomposition w.r.t. $\hat{p}_{t,i}$,

$$\mathbb{E}_{\substack{\theta^P\sim A^P \\ (X_t,Y_t)\sim\mathcal{D}}}[(p_{t,i} - \hat{p}_{t,i})^2]$$
$$= \mathbb{E}_{(X_t,Y_t)\sim\mathcal{D}}[\mathbb{E}_{\theta^P\sim A^P}[(p_{t,i} - \hat{p}_{t,i})^2]]$$
$$= \mathbb{E}_{(X_t,Y_t)\sim\mathcal{D}}[(p_{t,i} - \mathbb{E}_{\theta^P\sim A^P}[\hat{p}_{t,i}])^2 + \text{Var}_{\theta^P\sim A^P}[\hat{p}_{t,i}]]$$
$$= \mathbb{E}_{(X_t,Y_t)\sim\mathcal{D}}[(p_{t,i} - \mathbb{E}_{\theta^P\sim A^P}[\hat{p}_{t,i}])^2] + \mathbb{E}_{(X_t,Y_t)\sim\mathcal{D}}[\text{Var}_{\theta^P\sim A^P}[\hat{p}_{t,i}]]$$
$$= \epsilon_P^2 + \sigma_P^2.$$

Secondly, by the law of total expectation w.r.t. $(X_t, Y_t)$ and $\hat{p}_{t,i}$ and the bias-variance decomposition w.r.t. $f_{t,i}^C$,

$$\mathbb{E}_{\substack{\theta^P \sim A^P \\ \theta^C \sim A^C \\ (X_t, Y_t) \sim \mathcal{D}}} [(p_{t,i} - \hat{p}_{t,i} - f_{t,i}^C)^2]$$

$$= \mathbb{E}_{\substack{\theta^P \sim A^P \\ (X_t, Y_t) \sim \mathcal{D}}} [\mathbb{E}_{\theta^C \sim A^C}[(p_{t,i} - \hat{p}_{t,i} - f_{t,i}^C)^2]]$$

$$= \mathbb{E}_{\substack{\theta^P \sim A^P \\ (X_t, Y_t) \sim \mathcal{D}}} [(p_{t,i} - \hat{p}_{t,i} - \mathbb{E}_{\theta^C \sim A^C}[f_{t,i}^C])^2 + \mathrm{Var}_{\theta^C \sim A^C}[f_{t,i}^C]]$$

$$= \mathbb{E}_{\substack{\theta^P \sim A^P \\ (X_t, Y_t) \sim \mathcal{D}}} [(p_{t,i} - \hat{p}_{t,i} - \mathbb{E}_{\theta^C \sim A^C}[f_{t,i}^C])^2] + \mathbb{E}_{\substack{\theta^P \sim A^P \\ (X_t, Y_t) \sim \mathcal{D}}} [\mathrm{Var}_{\theta^C \sim A^C}[f_{t,i}^C]]$$

$$= \epsilon_C^2 + \sigma_C^2.$$

Thirdly, by the law of total expectation w.r.t. $(X_t, Y_t)$ and $\hat{p}_{t,i}$ and the Cauchy–Schwarz inequality,

$$\mathbb{E}_{\substack{\theta^P \sim A^P \\ \theta^C \sim A^C \\ (X_t, Y_t) \sim \mathcal{D}}} [(p_{t,i} - \hat{p}_{t,i})(p_{t,i} - \hat{p}_{t,i} - f_{t,i}^C)]$$

$$= \mathbb{E}_{\substack{\theta^P \sim A^P \\ (X_t, Y_t) \sim \mathcal{D}}} [\mathbb{E}_{\theta^C \sim A^C}[(p_{t,i} - \hat{p}_{t,i})(p_{t,i} - \hat{p}_{t,i} - f_{t,i}^C)]]$$

$$= \mathbb{E}_{\substack{\theta^P \sim A^P \\ (X_t, Y_t) \sim \mathcal{D}}} [(p_{t,i} - \hat{p}_{t,i})(p_{t,i} - \hat{p}_{t,i} - \mathbb{E}_{\theta^C \sim A^C}[f_{t,i}^C])]$$

$$\leq \sqrt{\mathbb{E}_{\substack{\theta^P \sim A^P \\ (X_t, Y_t) \sim \mathcal{D}}} [(p_{t,i} - \hat{p}_{t,i})^2] \cdot \mathbb{E}_{\substack{\theta^P \sim A^P \\ (X_t, Y_t) \sim \mathcal{D}}} [(p_{t,i} - \hat{p}_{t,i} - \mathbb{E}_{\theta^C \sim A^C}[f_{t,i}^C])^2]}$$

$$= \sqrt{(\epsilon_P^2 + \sigma_P^2) \cdot \epsilon_C^2} = \epsilon_C \sqrt{\epsilon_P^2 + \sigma_P^2}.$$

Together, as long as $0 \leq \lambda \leq 1$,

$$\mathbb{E}_{\substack{\theta^P \sim A^P \\ \theta^C \sim A^C \\ (X_t, Y_t) \sim \mathcal{D}}} [(p_{t,i} - \hat{p}_{t,i} - \lambda f_{t,i}^C)^2] - \mathbb{E}_{\substack{\theta^P \sim A^P \\ (X_t, Y_t) \sim \mathcal{D}}} [(p_{t,i} - \hat{p}_{t,i})^2]$$

$$= \mathbb{E}_{\substack{\theta^P \sim A^P \\ \theta^C \sim A^C \\ (X_t, Y_t) \sim \mathcal{D}}} [((1-\lambda)(p_{t,i} - \hat{p}_{t,i}) + \lambda(p_{t,i} - \hat{p}_{t,i} - f_{t,i}^C))^2] - \mathbb{E}[(p_{t,i} - \hat{p}_{t,i})^2]$$

$$= (1-\lambda)^2 \mathbb{E}_{\substack{\theta^P \sim A^P \\ \theta^C \sim A^C \\ (X_t, Y_t) \sim \mathcal{D}}} [(p_{t,i} - \hat{p}_{t,i})^2] + \lambda^2 \mathbb{E}_{\substack{\theta^P \sim A^P \\ \theta^C \sim A^C \\ (X_t, Y_t) \sim \mathcal{D}}} [(p_{t,i} - \hat{p}_{t,i} - f_{t,i}^C)^2]$$

$$+ 2(1-\lambda)\lambda \mathbb{E}_{\substack{\theta^P \sim A^P \\ \theta^C \sim A^C \\ (X_t, Y_t) \sim \mathcal{D}}} [(p_{t,i} - \hat{p}_{t,i})(p_{t,i} - \hat{p}_{t,i} - f_{t,i}^C)] - \mathbb{E}_{\substack{\theta^P \sim A^P \\ (X_t, Y_t) \sim \mathcal{D}}} [(p_{t,i} - \hat{p}_{t,i})^2]$$

$$= (1-\lambda)^2 \mathbb{E}_{\substack{\theta^P \sim A^P \\ (X_t, Y_t) \sim \mathcal{D}}} [(p_{t,i} - \hat{p}_{t,i})^2] + \lambda^2 \mathbb{E}_{\substack{\theta^P \sim A^P \\ \theta^C \sim A^C \\ (X_t, Y_t) \sim \mathcal{D}}} [(p_{t,i} - \hat{p}_{t,i} - f_{t,i}^C)^2]$$

$$+ 2(1-\lambda)\lambda \mathbb{E}_{\substack{\theta^P \sim A^P \\ \theta^C \sim A^C \\ (X_t, Y_t) \sim \mathcal{D}}} [(p_{t,i} - \hat{p}_{t,i})(p_{t,i} - \hat{p}_{t,i} - f_{t,i}^C)] - \mathbb{E}_{\substack{\theta^P \sim A^P \\ (X_t, Y_t) \sim \mathcal{D}}} [(p_{t,i} - \hat{p}_{t,i})^2]$$

$$= ((1-\lambda)^2 - 1) \mathbb{E}_{\substack{\theta^P \sim A^P \\ (X_t, Y_t) \sim \mathcal{D}}} [(p_{t,i} - \hat{p}_{t,i})^2] + \lambda^2 \mathbb{E}_{\substack{\theta^P \sim A^P \\ \theta^C \sim A^C \\ (X_t, Y_t) \sim \mathcal{D}}} [(p_{t,i} - \hat{p}_{t,i} - f_{t,i}^C)^2]$$

$$+ 2(1-\lambda)\lambda \mathbb{E}_{\substack{\theta^P \sim A^P \\ \theta^C \sim A^C \\ (X_t, Y_t) \sim \mathcal{D}}} [(p_{t,i} - \hat{p}_{t,i})(p_{t,i} - \hat{p}_{t,i} - f_{t,i}^C)]$$

$$\leq ((1-\lambda)^2 - 1)(\epsilon_P^2 + \sigma_P^2) + \lambda^2(\epsilon_C^2 + \sigma_C^2) + 2(1-\lambda)\lambda \epsilon_C \sqrt{\epsilon_P^2 + \sigma_P^2}$$

$$= \lambda\left(\left(\left(\sqrt{\epsilon_P^2 + \sigma_P^2} - \epsilon_C\right)^2 + \sigma_C^2\right)\lambda - 2\sqrt{\epsilon_P^2 + \sigma_P^2}\left(\sqrt{\epsilon_P^2 + \sigma_P^2} - \epsilon_C\right)\right),$$

which is strictly smaller than 0 as long as

$$0 < \lambda < \min\left\{1, \frac{2\sqrt{\epsilon_P^2 + \sigma_P^2}\left(\sqrt{\epsilon_P^2 + \sigma_P^2} - \epsilon_C\right)}{\left(\sqrt{\epsilon_P^2 + \sigma_P^2} - \epsilon_C\right)^2 + \sigma_C^2}\right\} =: \lambda_0. \qquad \square$$

# C    Additional Empirical Analysis

**Question:** Zach wants to ride the Ferris wheel , the roller coaster , and the log ride . The Ferris wheel costs 2 tickets , the roller coaster costs 7 tickets, and the log ride costs 1 ticket . Zach has 1 ticket . How many more tickets should Zach buy ?

**Pilot Model Only Answer**

To find out how many tickets Zach needs to buy, we need to add up the cost of each ride and subtract the number of tickets Zach already has. - The Ferris wheel costs 2 tickets. - The roller coaster costs 7 tickets. - The log ride costs 1 ticket. Adding these up, we get: 2 + 7 + 1 = 10. So, Zach needs to buy **10 - 1 = 10 tickets**. Therefore, Zach should buy **10** tickets.

**Pilot Model *with* T-Copilot Answer**

To find out how many tickets Zach needs to buy, we need to add up the cost of each ride and subtract the number of tickets Zach already has. - The Ferris wheel costs 2 tickets. - The roller coaster costs 7 tickets. - The log ride costs 1 ticket. Adding these up, we get: 2 + 7 + 1 = 10. So, Zach needs to buy **10 - 1 = 9 tickets**. Therefore, Zach should buy **9** tickets.

Figure 7: Example of Copilot's Token-level Rectification on MAWPS.

**Setups.** In our empirical analysis, we choose LLaMA-3.2-3B as the Pilot model and T-Copilot-1B as the Copilot model. The Copilot model's implementation details are the same as stated in Appendix E.2. We evaluate on two reasoning tasks, including SIQA [70] and MAWPS [42]. The dataset details are provided later in Appendix D.

**Example of Copilot's Token-level Rectification.** Figure 7 demonstrates another representative example of Copilot's token-level rectification on the factual error made by the Pilot model. The token "10" is originally predicted wrong during the Pilot model mid-generation and is later corrected (token "9") through the Copilot model's logits rectification. To visualize the process, we present three plots showing the top-5 tokens' output logits and probabilities in the current token prediction. Note that the Copilot not only increases the logits value on the groundtruth token but also decreases the logits value on the original Pilot model's falsely predicted token. We further apply the Logit Lens [8], a standard interpretability tool, to project hidden state embeddings from each intermediate layer onto the vocabulary space to show how the Copilot adjusts the Pilot model's predictions on each state.

# D    Datasets

## D.1    Commonsense Reasoning

For the commonsense reasoning tasks, we choose six open-ended multiple-choice QA tasks. The detailed description for each dataset is listed below:

- **PIQA** [10]: A dataset for physical commonsense reasoning, requiring models to choose the more plausible solution for everyday tasks.
- **WinoGrande (WinoG.)** [68]: A large-scale dataset for commonsense pronoun resolution, extending the Winograd Schema Challenge [43] with diverse and harder examples.
- **HellaSwag (HellaS.)** [93]: A benchmark testing commonsense reasoning in story completion by selecting the most plausible next sentence among adversarial choices.
- **BoolQ** [18]: A question-answering dataset where models answer yes/no questions based on a given passage, requiring deep reading comprehension.
- **SIQA** [70]: A dataset for reasoning about social and emotional situations by selecting the most appropriate response to everyday scenarios.
- **Openbook QA (OBQA)** [55]: A dataset that tests knowledge-based question answering by requiring models to combine common knowledge with reasoning over multiple facts.

In our commonsense reasoning experiments, we follow the experimental setup from [33] and fine-tune both our models and baseline models on the combined training dataset, Commonsense170K, which is constructed by sampling and integrating the training sets of the aforementioned commonsense reasoning datasets. Each dataset's individual test set is used for evaluation. Both fine-tuning and testing data instances utilize zero-shot input prompts.

## D.2 Arithmetic Reasoning

For arithmetic reasoning tasks, we evaluate our method on four open-ended math problem-solving datasets spanning multiple mathematical domains. The detailed description of each dataset is provided below:

- **AQuA** [48]: A dataset of algebraic and arithmetic word problems presented in a multiple-choice format, requiring logical reasoning and numerical computation.
- **GSM8K** [19]: A dataset of grade-school-level math word problems designed to evaluate step-by-step reasoning and arithmetic skills.
- **MAWPS** [42]: A dataset aggregating math word problems from various sources, focusing on problem diversity and automatic equation generation.
- **SWAMP** [60]: A dataset that introduces systematic variations of simple arithmetic word problems to assess model robustness against linguistic perturbations

In our arithmetic reasoning experiments, we follow the experimental setup from [33] and fine-tune both our models and baseline model on the combined training dataset, Math10K. We also adopt the data preprocessing setup in [85] to avoid any potential training data leakage. Each aforementioned dataset's individual test set is used for evaluation. **Note that**, unlike commonsense reasoning, fine-tuning for arithmetic reasoning involves labels with zero-shot Chain-of-Thought (CoT) [41] prompts. Consequently, the training cutoff length is longer due to the increased token count and additional information contained in the prompts.

## D.3 Downstream tasks: Recommendation

For downstream application experiments, we utilize two sequential recommendation datasets, as LLM-based recommendation is a widely adopted task to evaluate language models' generation and decision-making capabilities. The detailed description for each dataset is listed below:

- **Beauty** [30]: The Beauty dataset comprises user-item interaction data from the Amazon beauty product category. It includes 22,363 users and 12,101 items, with a total of 198,502 interactions. The dataset has a sparsity level of 99.93%.
- **LastFM** [67]: The LastFM dataset contains 1,090 users and 3,646 items, with 52,551 interactions in total. The sparsity of the dataset is 98.68%.

In our experiments, we use the training and testing datasets from [88]. To ensure a fair comparison, we assign random numeric IDs to items and evaluate our method and baselines on sequential recommendation tasks.

**Metrics.** For evaluation, we employ two commonly used metrics Hit@$K$ and NDCG@$K$ metrics with $K \in \{5, 10, 20, 100\}$. We define each metric in detail below:

- **Hit Rate** measures the proportion of users for whom at least one relevant item appears within the top $K$ recommendations.

$$\text{H@}K = \frac{1}{|U|} \sum_{u \in U} \mathbb{I}(\text{Rel}(u) \cap R_u^K \neq \emptyset) \tag{9}$$

where $U$ is the set of users, $R_u^K$ is the top-K recommended items for user $u$, $\text{Rel}(u)$ is the set of relevant items for user $u$, and $\mathbb{I}(\cdot)$ is the indicator function which equals 1 if the condition is true, and 0 otherwise.

- **NDCG** evaluates both the relevance and position of items in the ranked list, assigning higher importance to relevant items that appear earlier, thereby reflecting the overall quality of the ranking

system.

$$\text{NDCG@}K = \frac{1}{|U|} \sum_{u \in U} \frac{\sum_{i=1}^{K} \frac{\text{rel}_{u,i}}{\log_2(i+1)}}{\sum_{i=1}^{|R_u^*|} \frac{\text{rel}_{u,i}}{\log_2(i+1)}} \tag{10}$$

where $\text{rel}_{u,i}$ is the relevance score of the item at position $i$ in the ranked list for user $u$, and $R_u^*$ is the ideal ranking of relevant items for user $u$.

## D.4 Fine-tuning Dataset Template

Table 4: Examples of dataset templates used in Transformer Copilot.

| Task Type | Template |
|---|---|
| **Commonsense Reasoning** | **Input:** Please choose the correct answer to fill in the blank to complete the given sentence: "After being diagnosed with Parkinson's, he now was taking supplements and prescriptions, but the doctor warned the ___ would be ineffective." Option1: supplements Option2: prescriptions Answer format: option1/option2

**Answer:** The correct answer is option1. |
| | **Input:** Please choose the correct ending to complete the given sentence: High jump: The boy lands on his back on to a red mat. The boy gets up from the mat. The boy: Ending1: starts doing spins. Ending2: celebrates by clapping and flexing both arms. Ending3: is dancing on the mat. Ending4: does jump jacks on his stick. Answer format: ending1/ending2/ending3/ending4

**Answer:** ending2 |
| | **Input:** Please answer the following question with true or false, question: is elder scrolls online the same as skyrim? Answer format: true/false.

**Answer:** false |
| **Arithmetic Reasoning** | **Input:** Please answer the following math question step by step: Darnel sprinted 0.875 of a lap and then took a break by jogging 0.75 of a lap. How much farther did Darnel sprint than jog?

**Answer:** Step 1: Find the total distance Darnel covered. Total distance = distance sprinted + distance jogged Total distance = 0.875 + 0.75 = 1.625 laps Step 2: Find the distance Darnel sprinted. Distance sprinted = 0.875 laps Step 3: Find the distance Darnel jogged. Distance jogged = 0.75 laps Step 4: Find the difference between the distance Darnel sprinted and jogged. Difference = Distance sprinted - Distance jogged Difference = 0.875 - 0.75 = 0.125 laps Therefore, Darnel sprinted 0.125 laps farther than jogged. The answer in Arabic numerals is 0.125. |
| | **Input:** If the probability that Stock A will increase in value during the next month is 0.56, and the probability that Stock B will increase in value during the next month is 0.74. What is the greatest value for the probability that neither of these two events will occur?

**Answer:** The probability that stock A does not increase is 0.44, and the probability that stock B does not increase is 0.26. Now, how can the probability that both do not increase be more than individual probability of not increasing for each? So the probability that both do not increase can not be more than 0.26. Basically the probability that both do not increase is between 0 and 0.26. |
| | **Input:** A typical tournament of tennis has 4 rounds. There are 8 games in the first round, 4 in the second round, 2 in the third round and 1 during the finals. If each game requires new tennis balls, and on average each game uses 5 cans of tennis balls, how many tennis balls in total are used at the end of the tournament if each can has 3 tennis balls?

**Answer:** Step 1: Calculate the number of tennis balls used in each game. Each can has 3 tennis balls, so 5 cans have 3 x 5 = 15 tennis balls. Therefore, each game uses 15 tennis balls. Step 2: Calculate the total number of games in the tournament. The first round has 8 games, the second round has 4 games, the third round has 2 games, and the finals have 1 game. Therefore, the total number of games in the tournament is 8 + 4 + 2 + 1 = 15.Step 3: Calculate the total number of tennis balls used in the tournament. Each game uses 15 tennis balls, so 15 games use 15 x 15 = 225 tennis balls.Therefore, the total number of tennis balls used in the tournament is 225.0. |
| **Downstream Recommendation** | **Input:** Considering {*dataset*}, user_{*user_id*} has interacted with {*dataset*} items {*history*}. What is the next recommendation for the user?

**Answer:** {*dataset*} {*target*}
E.g. Beauty item_1253 |

In Table 4, we provide examples of data instances for each task mentioned above during model fine-tuning. All experiments are conducted in the zero-shot setting to better facilitate model-wise evaluation using pass@1 accuracy (i.e., based on a single generation attempt).

# E  Experiment Setups

## E.1  Hyperparameters and Training/Inference Details

Tables 5-10 present our hyperparameter settings of each task for reproducibility. We perform hyperparameter tuning for both T-Copilot and baseline methods. Unless otherwise specified, both our method and baseline implementations use beam search decoding [23] during inference. All experiments have been run three times with random seeds, reporting average accuracy. For FLAN-T5, LLaMA-3, and Qwen2.5 models, checkpoints are saved every 1,000 steps to track parameters and monitor training to ensure robustness and avoid overfitting.

Table 5: Hyperparameter configuration of Transformer Copilot for LLaMA-3 and Qwen-2.5 series models on the **Commonsense Reasoning** Tasks.

| Hyperparameters | Pilot Model | | | Copilot Model |
|---|---|---|---|---|
| | LLaMA-3.2-1B | LLaMA-3.2-3B | LLaMA-3.1-8B | T-Copilot (1B) |
| $\lambda$ | [0.1, 0.3, 0.5, 0.8, 1.0] | | | |
| *Fine-tuning Configurations* | | | | |
| Epochs | 3 | 3 | 3 | 3 |
| Batch Size | 16 | 16 | 16 | 16 |
| Micro Batch Size | 4 | 4 | 4 | 4 |
| Cut Off Length | 256 | 256 | 256 | 256 |
| Maximum Learning Rate | $3e^{-4}$ | $3e^{-4}$ | $3e^{-4}$ | $5e^{-4}$ |
| Learning Rate Scheduler | Cosine | Cosine | Cosine | Cosine |
| Optimizer | AdamW | AdamW | AdamW | AdamW |
| Warmup Steps | 200 | 200 | 200 | 200 |
| Weight Decay | 0.00 | 0.00 | 0.00 | 0.00 |
| *LoRA Configurations* | | | | |
| Rank $r$ | 32 | 32 | 32 | 32 |
| LoRA Alpha | 64 | 64 | 64 | 64 |
| LoRA Dropout | 0.05 | 0.05 | 0.05 | 0.08 |
| *Inference Configurations* | | | | |
| Temperature | 0.1 | | | |
| Top p | 0.95 | | | |
| Top k | 40 | | | |
| Num Beams | 4 | | | |
| Maximum New Tokens | 64 | | | |

Table 6: Hyperparameter configuration of Transformer Copilot for LLaMA-3 and Qwen-2.5 series models on the **Arithemtic Reasoning** Tasks.

| Hyperparameters | Pilot Model | | | Copilot Model |
|---|---|---|---|---|
| | LLaMA-3.2-1B | LLaMA-3.2-3B | LLaMA-3.1-8B | T-Copilot (1B) |
| $\lambda$ | [0.1, 0.3, 0.5, 0.8, 1.0] | | | |
| *Fine-tuning Configurations* | | | | |
| Epochs | 3 | 3 | 3 | 3 |
| Batch Size | 16 | 16 | 16 | 16 |
| Micro Batch Size | 4 | 4 | 4 | 4 |
| Cut Off Length | 256 | 256 | 256 | 256 |
| Maximum Learning Rate | $2e^{-4}$ | $2e^{-4}$ | $1e^{-4}$ | $3e^{-4}$ |
| Learning Rate Scheduler | Cosine | Cosine | Cosine | Cosine |
| Optimizer | AdamW | AdamW | AdamW | AdamW |
| Warmup Steps | 100 | 100 | 100 | 100 |
| Weight Decay | 0.00 | 0.00 | 0.00 | 0.00 |
| *LoRA Configurations* | | | | |
| Rank $r$ | 32 | 32 | 32 | 32 |
| LoRA Alpha | 64 | 64 | 64 | 64 |
| LoRA Dropout | 0.05 | 0.05 | 0.05 | 0.08 |
| *Inference Configurations* | | | | |
| Temperature | 0.1 | | | |
| Top p | 0.95 | | | |
| Top k | 40 | | | |
| Num Beams | 4 | | | |
| Maximum New Tokens | 256 | | | |

Table 7: Hyperparameter configuration of Transformer Copilot for LLaMA-3.2-1B, LLaMA-3.2-3B, and LLaMA-3.1-8B on the **Downstream Recommendation** Tasks.

| Hyperparameters | Pilot Model | | | Copilot Model |
|---|---|---|---|---|
| | LLaMA-3.2-1B | LLaMA-3.2-3B | LLaMA-3.1-8B | T-Copilot (1B) |
| $\lambda$ | [0.1, 0.3, 0.5, 0.8, 1.0] | | | |
| *Fine-tuning Configurations* | | | | |
| Epochs | 3 | 3 | 3 | 3 |
| Batch Size | 32 | 32 | 32 | 32 |
| Micro Batch Size | 1 | 1 | 1 | 1 |
| Cut Off Length | 256 | 256 | 256 | 256 |
| Maximum Learning Rate | $3e^{-4}$ | $3e^{-4}$ | $3e^{-4}$ | $5e^{-4}$ |
| Learning Rate Scheduler | Cosine | Cosine | Cosine | Cosine |
| Optimizer | AdamW | AdamW | AdamW | AdamW |
| Warmup Steps | 100 | 100 | 100 | 100 |
| Weight Decay | 0.00 | 0.00 | 0.00 | 0.00 |
| *LoRA Configurations* | | | | |
| Rank $r$ | 16 | 16 | 16 | 16 |
| LoRA Alpha | 16 | 16 | 16 | 16 |
| LoRA Dropout | 0.05 | 0.05 | 0.05 | 0.08 |
| *Inference Configurations* | | | | |
| Temperature | 0.1 | | | |
| Top p | 0.95 | | | |
| Top k | 40 | | | |
| Num Beams | 4 | | | |
| Maximum New Tokens | 64 | | | |

Table 8: Hyperparameter configuration of Transformer Copilot for FLAN-T5-small/base/large on the **Commonsense Reasoning** Tasks.

| Hyperparameters | Pilot Model | | | Copilot Model |
|---|---|---|---|---|
| | FLAN-T5-small | FLAN-T5-base | FLAN-T5-large | T-Copilot (small/base) |
| $\lambda$ | [0.1, 0.3, 0.5, 0.8, 1.0] | | | |
| *Fine-tuning Configurations* | | | | |
| Epochs | 12 | 12 | 12 | 12 |
| Batch Size | 16 | 16 | 16 | 16 |
| Micro Batch Size | 1 | 1 | 1 | 1 |
| Cut Off Length | 256 | 256 | 256 | 256 |
| Maximum Learning Rate | $1e^{-3}$ | $1e^{-3}$ | $1e^{-3}$ | $3e^{-3}$ |
| Learning Rate Scheduler | Cosine | Cosine | Cosine | Cosine |
| Optimizer | AdamW | AdamW | AdamW | AdamW |
| Warmup Ratio | 0.05 | 0.05 | 0.05 | 0.05 |
| Weight Decay | 0.01 | 0.01 | 0.01 | 0.01 |
| Drop Out | 0.1 | 0.1 | 0.1 | 0.1 |
| *Inference Configurations* | | | | |
| Temperature | 0.1 | | | |
| Top p | 0.95 | | | |
| Top k | 40 | | | |
| Num Beams | 4 | | | |
| Maximum New Tokens | 64 | | | |

Table 9: Hyperparameter configuration of Transformer Copilot for FLAN-T5-small/base/large on the **Arithmetic Reasoning** Tasks.

| Hyperparameters | Pilot Model | | | Copilot Model |
|---|---|---|---|---|
| | FLAN-T5-small | FLAN-T5-base | FLAN-T5-large | T-Copilot (small/base) |
| $\lambda$ | [0.1, 0.3, 0.5, 0.8, 1.0] | | | |
| *Fine-tuning Configurations* | | | | |
| Epochs | 12 | 12 | 12 | 12 |
| Batch Size | 16 | 16 | 16 | 16 |
| Micro Batch Size | 1 | 1 | 1 | 1 |
| Cut Off Length | 256 | 256 | 256 | 256 |
| Maximum Learning Rate | $1e^{-3}$ | $1e^{-3}$ | $1e^{-3}$ | $3e^{-3}$ |
| Learning Rate Scheduler | Cosine | Cosine | Cosine | Cosine |
| Optimizer | AdamW | AdamW | AdamW | AdamW |
| Warmup Ratio | 0.05 | 0.05 | 0.05 | 0.05 |
| Weight Decay | 0.01 | 0.01 | 0.01 | 0.01 |
| Drop Out | 0.1 | 0.1 | 0.1 | 0.1 |
| *Inference Configurations* | | | | |
| Temperature | 0.1 | | | |
| Top p | 0.95 | | | |
| Top k | 40 | | | |
| Num Beams | 4 | | | |
| Maximum New Tokens | 256 | | | |

Table 10: Hyperparameter configuration of Transformer Copilot for T5-small/base on the **Downstream Recommendation** Tasks.

| Hyperparameters | Pilot Model | | Copilot Model |
|---|---|---|---|
| | T5-small | T5-base | T-Copilot (small/base) |
| $\lambda$ | [0.1, 0.3, 0.5, 0.8, 1.0] | | |
| *Fine-tuning Configurations* | | | |
| Epochs | 20 | 20 | 20 |
| Batch Size | 16 | 16 | 16 |
| Micro Batch Size | 1 | 1 | 1 |
| Cut Off Length | 256 | 256 | 256 |
| Maximum Learning Rate | $1e^{-3}$ | $1e^{-3}$ | $1e^{-3}$ |
| Learning Rate Scheduler | Cosine | Cosine | Cosine |
| Optimizer | AdamW | AdamW | AdamW |
| Warmup Ratio | 0.05 | 0.05 | 0.05 |
| Weight Decay | 0.01 | 0.01 | 0.01 |
| Drop Out | 0.1 | 0.1 | 0.1 |
| *Inference Configurations* | | | |
| Temperature | 0.1 | | |
| Top p | 0.95 | | |
| Top k | 40 | | |
| Num Beams | 4 | | |
| Maximum New Tokens | 64 | | |

Table 11: Total and Trainable Parameter Statistics. We report the total trainable parameter count for encoder-decoder models. For other model types, we present the proportion of trainable parameters under LoRA fine-tuning relative to the total model size.

| Type | Model | Size (Total) | Params (Trainable) |
|---|---|---|---|
| T5/FLAN-T5 | T5-small | 61M | 61M |
| | **+ T-Copilot-small** | 92M | 92M |
| | T5-small$_{12}$ | 122M | 122M |
| | T5-base | 223M | 223M |
| | **+ T-Copilot-base** | 349M | 349M |
| | T5-base$_{24}$ | 446M | 446M |
| | FLAN-T5-small | 77M | 77M |
| | **+ T-Copilot-small** | 118M | 118M |
| | FLAN-T5-base | 248M | 248M |
| | **+ T-Copilot-base** | 385M | 385M |
| | FLAN-T5-large | 783M | 783M |
| | **+ T-Copilot-small** | 824M | 824M |
| | **+ T-Copilot-base** | 920M | 920M |
| LLaMA Pro | Llama-Pro-8B | 8.9B | 0.832% |
| | Mistral-Pro-8B | 8.3B | 0.858% |
| MoE | Mistral-7B | 7.3B | 0.721% |
| | Ministral-8B | 8.0B | 0.821% |
| MergeKit | MergeKit-9B | 8.9B | 0.710% |
| Gemma | Gemma-2-9B | 9.2B | 0.813% |
| LLaMA | LLaMA-3.2-1B | 1.3B | 1.215% |
| | **+ T-Copilot-1B** | 2.4B | 1.246% |
| | LLaMA-3.2-3B | 3.2B | 1.018% |
| | **+ T-Copilot-1B** | 4.3B | 1.018% |
| | LLaMA-3.1-8B | 8.0B | 0.700% |
| | **+ T-Copilot-1B** | 9.1B | 0.705% |
| Qwen | Qwen2.5-3B | 3.1B | 1.244% |
| | **+ T-Copilot-0.5B** | 3.6B | 1.650% |
| | **+ T-Copilot-3B** | 6.1B | 1.263% |
| | Qwen2.5-7B | 7.6B | 0.814% |
| | **+ T-Copilot-0.5B** | 8.0B | 0.819% |
| | **+ T-Copilot-3B** | 10.8B | 0.815% |
| | Qwen2.5-14B | 14.8B | 0.211% |

## E.2 T-Copilot Configurations and Implementations

In our implementation, we integrate the Transformer Copilot learning framework into both encoder-decoder and decoder-only LLMs mentioned above. Specifically, we introduce a Copilot model as an auxiliary component to the original Transformer architecture. Below, we provide details on our models' implementation and notations.

**T5/FLAN-T5:**

- **T-Copilot-small:** This refers to our Copilot model being initialized from the decoder module of a pre-trained T5-small or FLAN-T5-small model. Specifically, T-Copilot-small consists of 6 decoder layers with a hidden state dimension of 512, 8-headed attention, and a logit distribution dimensionality of 32,100. To adopt the model for our method, we exclude the conventional positional embedding mechanism and omit the softmax layer typically used for normalizing logits

into probability distributions. Additionally, we add a linear layer to map the Copilot inputs from the logits distribution dimension to the decoder hidden state dimension. If the Copilot's hidden state dimension differs from the Pilot model, an additional linear layer is added for dimension alignment.

- **T-Copilot-base:** This refers to our Copilot model being initialized from the decoder module of a pre-trained T5-base or FLAN-T5-base model. The overall model implementation is similar to T-Copilot-small. T-Copilot-base consists of 12 decoder blocks with a hidden state dimension of 768, 12-headed attention, and a logits distribution dimensionality of 32,100.

**LLaMA-3:**

- **T-Copilot-1B:** This refers to our Copilot model being initialized from the decoder module of a pre-trained LLaMA-3.2-1B model. T-Copilot-1B consists of 16 decoder blocks with a hidden state dimension of 2048, 32-headed attention, and a logits distribution dimensionality of 128,256. To adapt the model for our method, we exclude the conventional positional embedding mechanism and omit the softmax layer typically used for normalizing logits into probability distributions. To accelerate training, we incorporate the flash-attention mechanism. To enhance inference efficiency, we apply mean pooling to the concatenated input hidden states $h_{t,i}(X_t; \theta_{t-1}^1)$ without compromising performance accuracy. We add a linear layer to map the Copilot inputs from the logits distribution dimension to the decoder hidden state dimension. If the Copilot's hidden state dimension differs from the Pilot model, an additional linear layer is added for dimension alignment.

- **T-Copilot-3B:** This refers to our Copilot model being initialized from the decoder module of a pre-trained LLaMA-3.2-3B. T-Copilot-1B consists of 28 decoder blocks with a hidden state dimension of 3072, 24-headed attention, and a logits distribution dimensionality of 128,256.

**Qwen2.5:** The model configurations for Qwen2.5 are similar to LLaMA-3 models as they share similar model implementation details. We provide the additional model configurations below:

- **T-Copilot-0.5B:** This refers to our Copilot model being initialized from the decoder module of a pre-trained Qwen2.5-0.5B. T-Copilot-0.5B consists of 24 decoder blocks with a hidden state dimension of 896, 14-headed attention, and a logits distribution dimensionality of 151,936.

- **T-Copilot-3B:** This refers to our Copilot model being initialized from the decoder module of a pre-trained Qwen2.5-3B. T-Copilot-3B consists of 36 decoder blocks with a hidden state dimension of 2048, 16-headed attention, and a logits distribution dimensionality of 151,936.

**Notation.** In our experiments, we represent our methods using the original model name "+" the Copilot model. For example, FLAN-T5-small+T-Copilot-small denotes the integration of FLAN-T5-small with T-Copilot-small, and LLaMA-3.1-8B+T-Copilot-1B indicates the incorporation of LLaMA-3.1-8B with T-Copilot-1B.

### E.3   Baseline Details

**Frontier Models.** Below, we detail the specific model versions of the backbone and baseline models in our experiments.

*(i) Encoder-Decoder Models:* We use T5 and FLAN-T5 [65] with different sizes as our backbone and baseline models for the encoder-decoder Transformer architecture: `T5-small`, `T5-base`, `T5-large` and `FLAN-T5-small`, `FLAN-T5-base`, `FLAN-T5-large`.

*(ii) Decoder-Only Models:* For the decoder-only models, we utilize the LLaMA-3 family [21] as our backbone and baseline models. Our experiments include `LLaMA-3.2-1B`, `LLaMA-3.2-3B`, `LLaMA-3.1-8B`, and `LLaMA-2-13B`.

*(iii) MoE Models:* For the Mixture-of-Expert based models, we use `Mistral-7B` with version `Mistral-7B-v0.3` and `Ministral-8B` with version `Ministral-8B-Instruct-2410`.

**Layer/Adapter Expansion Models.** In our experiments, we also compare against baseline methods that utilize layer and adapter expansion approaches. Below, we provide the model configurations and implementation details for these baselines.

*(i) LLaMA Pro [84]:* `LLaMA-Pro-8B` incorporates a content-addressable working memory module to store and retrieve task-relevant information. In our implementation, we initialized with the

LLaMA-3.1-8B base model and expanded the number of blocks from 32 to 40 using an interleaved approach. `Mistra-Pro-8B` is an enhanced version of the original Mistral model [37], augmented with additional Transformer blocks. The model excels in integrating general language understanding with domain-specific knowledge and follows the same methodology as LLaMA-Pro-8B for block expansion. Following [84], we use the version of Mistral-Pro-8B-v0.1.

*(ii) MergeKit [26]:* MergeKit is an open-source toolkit designed for efficiently merging LLM checkpoints to combine their strengths without additional training. In our experiments, we train and apply one MergeKit model named `MergeKit-9B`. `MergeKit-9B` is initialized from LLaMa-3.1-8B and replicates additional layers with post-merge healing. The model is merged using the Passthrough method. In our experiments, we first compare the model with the original LLaMA-3.1-8B to ensure that the merged model does not lead to performance degradation.

*(iii) TIES [89]:* `T5-small`$_{12}$ and `T5-base`$_{24}$ are T5 type models merged using the TIES method. `T5-small`$_{12}$ merges two T5-small models and extends the original T5-small to 12 encoder and decoder layers. And `T5-base`$_{24}$ merges two T5-base models and extends the original T5-base to 24 encoder and decoder layers by duplicating existing layers.

**Model Parameters.** In table 11, we provided the detailed model sizes and trainable parameters for both Transformer Copilot and baseline models.

# F    Additional Experiments

## F.1    Full Table Report on Baseline Comparison

Table 12: Full performance comparison (%) with frontier baselines **under matched-parameter scales**. Results are averaged over 3 independent runs.

| Model | Params | Commonsense Reasoning (Acc. ↑) | | | | | | | Arithmetic Reasoning (Acc. ↑) | | | | |
| --- | --- | --- | --- | --- | --- | --- | --- | --- | --- | --- | --- | --- | --- |
| | | PIQA | WinoG. | HellaS. | BoolQ | SIQA | OBQA | Avg. | AQuA | GSM8K | MAWPS | SVAMP | Avg. |
| *≤8B-level Frontier LLMs* | | | | | | | | | | | | | |
| Mistral-7B | 7B | 83.0 | 75.3 | 81.3 | 65.4 | 73.1 | 74.5 | 75.4 | 28.9 | 50.2 | 85.3 | 57.4 | 55.5 |
| LLaMA-Pro-8B | 8B | 88.4 | 81.4 | 86.9 | 73.9 | 76.1 | 77.8 | 80.8 | 38.2 | 57.2 | 92.5 | 63.5 | 62.9 |
| LLaMA-3.1-8B | 8B | 85.4 | 84.3 | 90.9 | 69.6 | 79.9 | 82.6 | 82.1 | 37.3 | 63.5 | 89.1 | 73.6 | 65.9 |
| Ministral-8B | 8B | 85.7 | 84.1 | 91.3 | 70.3 | 77.5 | 81.3 | 81.7 | 37.4 | 62.9 | 90.2 | 73.2 | 65.9 |
| Qwen2.5-3B + T-Copilot-0.5B | 3.5B | 85.4 | 79.1 | 91.3 | 66.8 | 78.1 | 86.0 | 81.1 | 57.3 | 74.2 | 91.8 | 82.8 | 76.5 |
| LLaMA-3.2-3B + T-Copilot-3B | 6B | 85.6 | 83.7 | 91.3 | 72.8 | 79.2 | 81.3 | 82.3 | 40.1 | 63.1 | 91.2 | 71.4 | 66.5 |
| Qwen2.5-3B + T-Copilot-3B | 6B | 87.8 | 81.7 | 94.0 | 68.7 | 79.9 | 89.4 | 83.6 | 59.4 | 76.8 | 92.6 | 83.5 | 78.1 |
| Qwen2.5-7B + T-Copilot-0.5B | 7.5B | 89.3 | 85.3 | 93.5 | 73.6 | 80.0 | 92.1 | 85.6 | 61.4 | 78.2 | 93.0 | 86.5 | 79.8 |
| *>8B-level Frontier LLMs* | | | | | | | | | | | | | |
| Gemma-2-9B | 9B | 81.4 | 82.8 | 93.5 | 70.2 | 79.5 | 86.1 | 82.3 | 40.1 | 64.3 | 82.7 | 75.0 | 65.5 |
| MergeKit-9B | 9B | 86.1 | 84.7 | 91.1 | 71.1 | 79.3 | 80.2 | 82.1 | 37.0 | 65.2 | 90.3 | 75.2 | 66.9 |
| Qwen2.5-14B | 14B | 91.8 | 85.6 | 94.3 | 75.2 | 84.5 | 93.1 | 87.4 | 63.5 | 79.5 | 92.4 | 87.9 | 80.8 |
| LLaMA-3.1-8B + T-Copilot-1B | 9B | 86.2 | 86.8 | 93.5 | 71.8 | 82.7 | 83.2 | 84.0 | 38.9 | 66.1 | 90.8 | 75.4 | 67.8 |
| Qwen2.5-7B + T-Copilot-3B | 10B | 92.5 | 87.2 | 95.3 | 74.8 | 84.3 | 94.9 | 88.2 | 64.2 | 79.7 | 94.8 | 88.1 | 81.7 |

Table 12 shows the full comparison results of T-Copilot against baseline models and methods with matched and larger parameter scales. Notably, under the same model architectures and with less pre-trained knowledge, LLaMA-3.2-3B+T-Copilot-3B outperforms LLaMA-3.1-8B with 2B fewer parameters, Qwen2.5-7B+T-Copilot-3B outperforms Qwen2.5-14B with 4B fewer parameters, and Qwen2.5-3B+T-Copilot-3B outperforms Qwen2.5-7B with 1B fewer parameters. Our method also outperforms other layer/adapter expansion baselines. These results underscore the parameter efficiency and architectural strength of our learning framework.

## F.2    Downstream Recommendation Evaluation

In Table 13 and Table 14, we report the results of T-Copilot on two downstream recommendation datasets: Beauty and LastFM. We choose T5 and LLaMA-3 series models as the backbone Pilot models. Overall, T-Copilot improves the Pilot models by an average of 16.6% across all evaluation metrics on the two datasets. Furthermore, compared to other baselines, incorporating T-Copilot

Table 13: Performance comparison on **Beauty**. All methods are evaluated using both Hit Rates (H@K) and Normalized Discounted Cumulative Gain (N@K). The performance gains are also reported relative to respective backbone methods.

| Models | Beauty | | | | | | | |
|---|---|---|---|---|---|---|---|---|
| | H@5 | H@10 | H@20 | H@100 | N@5 | N@10 | N@20 | N@100 |
| T5-small$_{12}$ | 1.9 | 3.2 | 5.3 | 15.4 | 1.3 | 1.8 | 3.9 | 6.2 |
| T5-small | 1.7 | 2.9 | 5.4 | 14.6 | 1.0 | 1.4 | 3.5 | 5.7 |
| **+ T-Copilot-small** | 2.4 (+0.7) | 3.4 (+0.5) | 6.2 (+0.8) | 17.8 (+3.2) | 1.6 (+0.6) | 2.1 (+0.7) | 4.5 (+1.0) | 6.4 (+0.7) |
| T5-base$_{24}$ | 2.6 | 4.6 | 7.5 | 18.6 | 2.3 | 2.9 | 4.7 | 6.8 |
| T5-base | 2.3 | 3.3 | 6.2 | 17.4 | 2.1 | 2.6 | 4.5 | 6.2 |
| **+ T-Copilot-base** | 3.2 (+0.9) | 4.4 (+1.1) | 8.2 (+2.0) | 19.8 (+2.4) | 2.7 (+0.6) | 3.3 (+0.7) | 5.2 (+0.7) | 7.2 (+1.0) |
| LLaMA-3.2-1B | 5.2 | 7.4 | 10.0 | 18.8 | 3.8 | 4.4 | 5.1 | 6.7 |
| **+ T-Copilot-1B** | 6.1 (+0.9) | 8.1 (+0.7) | 12.5 (+2.5) | 24.6 (+5.8) | 4.3 (+0.5) | 5.1 (+0.7) | 5.8 (+0.7) | 7.4 (+0.7) |
| LLaMA-3.2-3B | 5.1 | 7.6 | 10.8 | 22.1 | 3.6 | 4.5 | 5.3 | 7.2 |
| **+ T-Copilot-1B** | 6.7 (+1.6) | 8.6 (+1.0) | 13.2 (+2.4) | 25.6 (+3.5) | 4.3 (+0.7) | 5.6 (+1.1) | 5.9 (+0.6) | 7.8 (+0.6) |
| LLaMA-3.1-8B | 5.8 | 8.3 | 11.1 | 21.5 | 4.1 | 4.9 | 5.6 | 7.5 |
| **+ T-Copilot-1B** | 7.1 (+1.3) | 9.2 (+0.9) | 13.5 (+2.4) | 26.4 (+4.9) | 4.7 (+0.6) | 6.2 (+1.3) | 6.4 (+0.8) | 8.1 (+0.6) |

Table 14: Performance comparison on **LastFM**. All methods are evaluated using both Hit Rates (H@K) and Normalized Discounted Cumulative Gain (N@K). The performance gains are also reported relative to respective backbone methods.

| Models | LastFM | | | | | | | |
|---|---|---|---|---|---|---|---|---|
| | H@5 | H@10 | H@20 | H@100 | N@5 | N@10 | N@20 | N@100 |
| T5-small$_{12}$ | 2.5 | 3.8 | 4.9 | 12.4 | 1.8 | 2.2 | 2.8 | 3.9 |
| T5-small | 2.1 | 3.7 | 4.2 | 11.0 | 1.6 | 2.0 | 2.5 | 3.2 |
| **+ T-Copilot-small** | 3.2 (+1.1) | 4.4 (+0.7) | 5.7 (+1.5) | 15.3 (+4.3) | 1.9 (+0.3) | 3.2 (+1.2) | 3.8 (+1.3) | 4.0 (+0.8) |
| T5-base$_{24}$ | 3.8 | 4.6 | 7.1 | 17.5 | 2.0 | 3.8 | 3.3 | 4.7 |
| T5-base | 2.7 | 4.2 | 5.3 | 14.9 | 1.9 | 2.4 | 2.9 | 3.4 |
| **+ T-Copilot-base** | 4.2 (+1.5) | 5.1 (+0.9) | 8.1 (+2.8) | 19.4 (+4.5) | 2.3 (+0.4) | 3.5 (+1.1) | 4.2 (+1.3) | 5.2 (+1.8) |
| LLaMA-3.2-1B | 5.0 | 5.7 | 9.1 | 21.9 | 2.4 | 3.0 | 3.9 | 6.2 |
| **+ T-Copilot-1B** | 6.4 (+1.4) | 6.8 (+1.1) | 11.2 (+2.1) | 24.7 (+2.8) | 2.9 (+0.5) | 3.5 (+0.5) | 4.3 (+0.4) | 6.7 (+0.5) |
| LLaMA-3.2-3B | 6.1 | 6.4 | 9.2 | 23.9 | 2.6 | 3.5 | 4.2 | 6.8 |
| **+ T-Copilot-1B** | 6.8 (+0.7) | 7.4 (+1.0) | 12.1 (+2.9) | 25.1 (+1.2) | 3.1 (+0.5) | 4.2 (+0.7) | 5.3 (+1.1) | 7.5 (+0.7) |
| LLaMA-3.1-8B | 4.7 | 7.3 | 10.3 | 25.6 | 3.1 | 3.7 | 4.7 | 7.0 |
| **+ T-Copilot-1B** | 6.9 (+2.2) | 8.6 (+1.3) | 12.7 (+2.4) | 28.0 (+2.4) | 3.9 (+0.8) | 4.8 (+1.1) | 5.4 (+0.7) | 7.9 (+0.9) |

achieves 16.7% and 8.6% higher performance than T5-small$_{12}$ and T5-base$_{24}$, respectively, on Beauty and LastFM, while using 30M and 126M fewer parameters. These results demonstrate that the error-correction capabilities of T-Copilot are not confined to reasoning tasks but also generalize effectively to other application domains, such as recommendation, where precise LLM decision-making is critical for downstream utility.

### F.3 Efficiency Analysis on Transformer Copilot

Table 15 presents the inference latency evaluation across six reasoning datasets. Our learning framework achieves lower latency than baseline models with comparable parameter scales. Specifically,

Table 15: Efficiency Comparison on Inference Latency. We report the total response time (s) per instance across six commonsense reasoning datasets, along with the average result.

| Inference Latency (↓) | PIQA | WinoG. | HellaS. | BoolQ | SIQA | OBQA | Avg. |
|---|---|---|---|---|---|---|---|
| LLaMA-3.2-1B | 0.33 | 0.36 | 0.27 | 0.23 | 0.27 | 0.24 | 0.28 |
| **+ T-Copilot-1B** | 0.36 | 0.39 | 0.28 | 0.26 | 0.29 | 0.25 | 0.31 |
| LLaMA-3.2-3B | 0.46 | 0.45 | 0.47 | 0.46 | 0.46 | 0.55 | 0.48 |
| **+ T-Copilot-1B** | 0.48 | 0.47 | 0.49 | 0.46 | 0.48 | 0.56 | 0.49 |
| LLaMA-3.1-8B | 0.52 | 0.52 | 0.51 | 0.49 | 0.49 | 0.62 | 0.53 |
| **+ T-Copilot-1B** | 0.52 | 0.53 | 0.53 | 0.49 | 0.50 | 0.63 | 0.53 |
| LLaMA-Pro-8B | 0.83 | 0.75 | 0.82 | 0.76 | 0.75 | 0.73 | 0.77 |
| MergeKit-9B | 0.64 | 0.64 | 0.57 | 0.54 | 0.63 | 0.72 | 0.62 |

LLaMA-3.1-8B+T-Copilot-1B consistently achieves 22.9% lower inference latency, 3% higher training throughput, and 57% higher tokens-per-second (TPS) on average compared to methods such as LLaMA-Pro-8B and MergeKit-9B. Furthermore, we observe that incorporating T-Copilot increases the inference latency by less than 2% relative to the original Pilot models, while yielding significant performance gains.

### F.4 Ablations and Analyses on Transformer Copilot

In this section, we perform multiple ablation studies to evaluate the influence of key hyperparameters and alternative method design on the T-Copilot's overall performance.

**Model Design of T-Copilot.** Table 16 compares T-Copilot-1B with a variant that excludes learning from the Pilot model's intermediate fine-tuning stages. The superior performance of T-Copilot highlights the advantage of our joint training paradigm, where the Mistake Log is continuously updated throughout the Pilot's training trajectory and enables the Copilot to effectively leverage intermediate-stage information.

Table 16: Ablation study on model design. We denote *Latest* as the variant where the 1B Copilot is trained using only the latest Pilot checkpoint.

| Pilot | Copilot | AQuA | GSM8K | MAWPS | SVAMP | Avg. |
|---|---|---|---|---|---|---|
| LLaMA-3.2-1B | Latest | 27.5 | 30.1 | 79.4 | 49.6 | 46.7 |
| | **T-Copilot** | **28.3** | **32.2** | **81.5** | **51.6** | **48.4** |
| LLaMA-3.2-3B | Latest | 34.6 | 57.1 | 87.5 | 65.2 | 61.1 |
| | **T-Copilot** | **36.6** | **58.2** | **89.1** | **68.7** | **63.2** |
| LLaMA-3.1-8B | Latest | 37.6 | 64.6 | 90.0 | 73.9 | 66.5 |
| | **T-Copilot** | **38.9** | **66.1** | **90.8** | **75.4** | **67.8** |

Table 17: Ablation study on $\lambda$. We use the T-Copilot-1B on LLaMA-3 series models.

| $\lambda$ | HellaSwag | | | GSM8K | | |
|---|---|---|---|---|---|---|
| | 1B | 3B | 8B | 1B | 3B | 8B |
| 0.3 | 62.0 | 90.6 | 90.9 | 29.8 | 56.8 | 64.4 |
| 0.5 | 62.8 | 90.9 | 91.5 | 30.4 | 57.6 | 65.9 |
| 0.8 | 63.1 | **91.2** | 92.4 | 31.8 | 58.1 | 65.7 |
| 1.0 | **63.3** | 91.1 | **93.5** | **32.2** | **58.2** | **66.1** |

**Design Variants of the Decoder-Only Copilot.** To validate the efficacy of our proposed decoder-only Copilot design, we explore several alternative architectural variants and empirically compare their impact on the model's final performance. Specifically, we examine different insertion patterns for the Copilot's new attention mechanism, i.e., the input and hidden states representations from the Pilot model recorded in the Mistake Log. We experiment with various design patterns and modify the Decoder-only Copilot model accordingly. The design options are listed below:

- **Pattern 1 (Ours):** Collect the hidden states across all Pilot model layers $L^P$ and insert them as key-value (KV) inputs for the **even-numbered** layers of the Copilot model.
- **Pattern 2:** Collect the hidden states across all Pilot model layers $L^P$ and insert them as KV inputs for **each** layer of the Copilot model. This setup examines whether integrating hidden states into all layers of the Copilot model improves performance by leveraging more entry points for processing the pilot model's hidden states information.

- **Pattern 3:** Collect only the **first half** ($L^P/2$) layers of the Pilot model's hidden states and insert them as key-value (KV) inputs for the Copilot model. Combined with Pattern 4, this setup investigates where the Pilot model makes more mistakes during the learning trajectory.
- **Pattern 4:** In contrast to Pattern 3, Pattern 4 collects only the **second half** ($L_1$ layers) of the Pilot model's hidden states and inserts them as KV inputs for the Copilot model.

Table 18: Empirical comparison of different design patterns of the Decoder-only Copilot model. We evaluate the LLaMA-3.2-1B Pilot model and T-Copilot-1B. We report the average accuracy on three independent runs. The highest accuracy for each dataset is highlighted in bold.

| Input Patterns | PIQA | HellaSwag | BoolQ | AQuA | GSM8K | SWAMP |
|---|---|---|---|---|---|---|
| Pattern 1 | **80.2** | **63.3** | **65.5** | **28.3** | **32.2** | **51.6** |
| Pattern 2 | 78.4 | 61.2 | 62.8 | 27.1 | 27.9 | 49.3 |
| Pattern 3 | 75.7 | 60.7 | 63.6 | 28.1 | 30.4 | 49.8 |
| Pattern 4 | 79.3 | 63.1 | 63.6 | 27.2 | 31.8 | 50.4 |

We follow the same experiment setups as stated in Section 5. Table 18 compares all 4 patterns on three commonsense reasoning and three arithmetic reasoning tasks. The results of Pattern 2 indicate that without the self-attention mechanism to capture dependencies in the Copilot model's generated outputs, the Copilot model struggles to effectively leverage the additional hidden state information during fine-tuning and inference. Additionally, the results comparing Pattern 3 and Pattern 4 do not reveal a clear performance trend. This suggests that the Pilot model makes mistakes at different layers depending on the assigned task. Therefore, the Mistake Log $M_T$ should capture all hidden states from the Pilot model to ensure that no relevant error-related information is omitted during the Copilot model's learning process. Based on this empirical analysis, we demonstrate the effectiveness of Pattern 1 for our Copilot model design.

**Choice of $\lambda$.** In theorem 4.1, we theoretically provide a bound on the range of $\lambda$ with $0 < \lambda < \lambda_0$. Here, we empirically study the effect of different $\lambda$ configurations. The results in Table 17 show that performance generally improves with larger $\lambda$ values in the range $[0, 1]$. The optimal value is observed around $\lambda = 1.0$. The results demonstrate that higher $\lambda$ amplifies the effect of T-Copilot, which aligns with our Copilot model design.

Table 19: Comparison of pooling methods for LLaMA-3.1-8B + T-Copilot-1B.

| Pooling Methods | GSM8K | AQuA |
|---|---|---|
| Max Pooling | 63.2 | 37.5 |
| Sum Pooling | 65.8 | 38.3 |
| **Mean Pooling** | **66.1** | **38.9** |

Table 20: Comparison of loss functions for Qwen2.5-7B + T-Copilot-3B.

| Loss Type | PIQA | AQuA |
|---|---|---|
| CE | 88.1 | 60.7 |
| KL Divergence | 90.4 | 62.8 |
| **RMSE** | **92.5** | **64.2** |

**Pooling Methods of Copilot's Attention Input.** In Section 3, we refine the Copilot model's attention mechanism by incorporating the mean-pooled hidden representations from the Pilot model as inputs (Eq. 6). To assess the effect of this design choice, we also experiment with alternative pooling strategies, including max pooling and sum pooling. The comparative results are presented in Table 19. From the results, the mean pooling empirically provides the best balance of stability and efficiency.

**Loss Types of T-Copilot.** We adopt the RMSE loss as the training objective for T-Copilot. To validate this choice, we conduct ablations with alternative loss formulations, including Cross-Entropy and KL Divergence. The results, summarized in Table 20, show that RMSE consistently yields superior performance across benchmarks.

# G   Additional Related Works

**Transformers for Language Modeling** The Transformer is a sequence-to-sequence model architecture that employs attention-based mechanisms, making it highly effective for autoregressive language modeling [2, 13, 64]. The vanilla Transformer [77] follows an **encoder-decoder** structure, comprising a stack of identical layers in both the encoder and decoder components. Each layer consists of a multi-head self-attention mechanism [15], layer normalization [4], a position-wise feedforward network, and residual connection [29]. The encoder-decoder structure serves as the foundation for many early-stage influential LLMs, such as T5 [65], and BART [44]. These models have demonstrated strong capabilities on certain generation tasks [38, 66]. On the other hand, (causal) **decoder-only** Transformer models [62, 63], trained with the autoregressive language modeling objective [69, 78], have demonstrated exceptional performance in open-ended generation and reasoning tasks [13, 82, 81]. The superior generalization capabilities have established decoder-only Transformers as the backbone of recent state-of-the-art LLMs such as PaLM [16], Falcon [3], LLaMA [76, 21], and ChatGPT [1, 50, 35]. In this work, we develop the Transformer Copilot framework to support both encoder-decoder and decoder-only Transformer architectures. Our intuition is to provide flexibility across a broad range of model configurations and downstream task scenarios.

**LLMs Adaptation with Fine-tuning.** Large language models perform well across many NLP tasks [98, 92], yet adapting them to specialized tasks remains challenging [80]. The standard solution, full-parameter fine-tuning [51], retrains all model parameters on task-specific data. Applying full fine-tuning has proven effective at improving performance, but can sometimes be computationally costly [73]. Recent work on parameter-efficient fine-tuning approaches [39, 32, 86, 100, 28], such as prefix-tuning [46] and LoRA [32], aims to reduce the computational overhead by tuning only a small subset of model parameters while still leveraging the expressive power of pre-trained models. Our learning framework builds upon the aforementioned methods' fine-tuning paradigm and aims to refine the fine-tuning and inference by utilizing mistake information during the models' learning trajectory. Additionally, since the Copilot model retains the decoder module structure, our framework can seamlessly integrate with various adaptation techniques such as DoRA [49] and ReFT [85].

**Differences from Boosting and neural exploration**. The core idea of Boosting [9, 24] is to train a series of models, where each subsequent model focuses on correcting the errors made by the previous ones. However, the proposed Copilot framework is distinct from boosting in several key ways. First, in boosting, the subsequent model is trained to correct the errors of a fixed, post-trained weak model, whereas the Copilot learns from the mistakes (errors) made by a strong, pre-trained pilot model during its fine-tuning trajectory. Second, the labels (errors) that the subsequent model in boosting attempts to predict are derived from the fixed parameters of the preceding weak model, whereas the labels that the Copilot learns are based on the fine-tuning dynamics of the Pilot's parameters. Third, while all models in boosting only take data features as inputs, the Copilot also takes the internal state of the Pilot model as part of its input. Fourth, boosting does not require modifications to the base models, whereas the Copilot framework involves modifying the model structure, specifically the Transformer architecture. Another related work is neural exploration methods [36, 87, 61, 5, 7, 83]. For example, one recent work called EE-Net [6] introduces an exploration neural network to manage trade-offs between exploitation and exploration in the contextual bandits setting. In contrast, the Copilot focuses on learning from the mistakes of the Pilot model in an offline, supervised learning regime, specifically tailored for Transformer-based sequence-to-sequence generation tasks.

## H Broader impact and Limitation

**Broader Impact.**   This paper introduces Transformer Copilot, a novel framework that enhances LLM fine-tuning by introducing a Mistake Log and an auxiliary Copilot model that learns to rectify errors during inference. Our approach improves model reliability and efficiency with minimal overhead and promotes more transparent and interpretable behavior by grounding predictions in prior training dynamics. While our method has broad applicability across domains, we do not foresee any specific societal risks or negative impacts that require special consideration.

**Limitation.**   While Transformer Copilot demonstrates robust improvements in inference quality by leveraging model-internal training signals, one potential consideration for future work lies in the coverage and diversity of the Mistake Log itself. Since the Mistake Log is constructed from the forward pass during supervised fine-tuning, its quality is inherently dependent on the richness and representativeness of the fine-tuning data distribution. In scenarios with limited domain coverage or skewed data sources, the Mistake Log may capture a narrower set of error patterns, potentially limiting the Copilot's generalizability. On the other hand, in our primary experimental setup, we fine-tune on task-diverse datasets with ample coverage, which ensures that the Mistake Log remains informative and representative. Our transferability experiments on the Copilot model further validate the Mistake Log's utility across unseen Pilot models, suggesting robustness to architectural and distributional shifts. Still, exploring data augmentation strategies or adaptive logging policies to enrich Mistake Logs for low-resource or domain-shifted settings remains an interesting future direction.

Overall, Transformer Copilot offers a promising paradigm shift toward internal signal utilization during LLM fine-tuning. We are optimistic that future research will build upon these contributions to develop even more precise and generalizable models through the continued adoption of reflective learning mechanisms.

