# OpenReview forum: "Transformer Copilot: Learning from The Mistake Log in LLM Fine-tuning"
_NeurIPS.cc/2025/Conference — NeurIPS 2025 spotlight_

### Official Review · Reviewer_sdqu · 2025-06-08

**Clarity:** 3
**Significance:** 3
**Originality:** 3
**Rating:** 5
**Confidence:** 3

**Summary:**

This paper introduces a new fine-tuning paradigm, the Transformer Copilot that incorporates an additional Copilot model that learns from the base model's (termed Pilot model) mistakes saved in a Mistake Log. The Copilot model's objective is to correct the Pilot model's output logits to improve performance on down-stream tasks. There are extensive empirical analysis and evaluations provided to support the Transformer Copilot's success. Further, a simple theoretical analysis is provided to understand the efficacy of their method.

**Questions:**

- Is there any reason the Copilot model should be of the same backbone as the Pilot model? Could it be possible that training a different backbone from scratch, not fine-tuning, may lead to better results?
- Have other pooling methods (besides mean pooling) been tested?
- There seems to be a trend for increasing $\lambda$ values to improve performance. What would be the effect of increasing $\lambda > 1$?

**Ethical Concerns:**

["NO or VERY MINOR ethics concerns only"]

**Final Justification:**

My initial assessment of the paper was already quite positive and the authors have carefully considered and responded to my questions. Overall, I believe this paper is a novel approach to fine-tuning that will contribute well to the community.

**Limitations:**

Yes

**Quality:**

4

**Strengths And Weaknesses:**

Strengths:
- Well-written and structured paper, allowing for smooth reading and clear understanding of each section.
- A novel approach to learn how to correct a model's ouput logits during inference. The method is intuitive and simple to understand.
- Extensive experiments are conducted to validate the proposed methods successfully, particually appreciated Table 2 that compared performances with baselines of similar parameter count as this was a natural question that arose during the reading of the paper.
- Emprical evidence is also provided to validate that the Copilot model indeed does adjust the Pilot model's logits significantly, leading to a change in final prediction.

Weaknesses:
- Theoretical justifications only focus on specific dimensions to prove a range of $\lambda$ values that ensure improved performance. Is it possible to extend these to consider the vector as a whole? Since, each dimension $k$ may have a different range of $\lambda$ that are disjoint, this could lead to scenarious where an optimal $\lambda$ cannot be found. Though, the experimental results emprically validate the method sufficiently, this is more for theoretical curiousity.
- In Figure 4, instead of showcasing a simple example of how the final token prediction has changed, it might be stronger to include some metrics that show how often Copilot changes the final token prediction correctly and wrongly.

---

> ### Author Rebuttal · Authors · 2025-07-31
>
> We sincerely thank the reviewer for your extensive and thoughtful feedback! We will address each of your remaining concerns below.
>
> ---
> ## **[W.1] Theoretical Justification for $\lambda$**
> Thank you for this insightful question! Below, we justify both theoretically and experimentally.
>
> > "Is it possible to extend these to consider the vector as a whole?"
>
> **Theoretically**: Yes, Theorem 1 can be extended to establish that there exists a constant $\lambda$ such that the theoretical guarantee holds for all $K$ dimensions (where $K = |V|$). Specifically, recall that for each dimension $k \in [K]$, there exists a threshold $\lambda_0^k$ such that the guarantee holds for that dimension when $0 < \lambda < \lambda_0^k$, where
> $\lambda_0^k = \frac{2\sqrt{\epsilon_P^2+\sigma_P^2}\big(\sqrt{\epsilon_P^2+\sigma_P^2}-\epsilon_C\big)}{\big(\sqrt{\epsilon_P^2+\sigma_P^2}-\epsilon_C\big)^2+\sigma_C^2}$. Let $\widehat{\lambda}_0 = \min\{\lambda_0^k: k \in [K]\}$. Then, the guarantee holds uniformly across all $K$ dimensions whenever $0 < \lambda < \widehat{\lambda}_0$.
>
> **Empirically**: It is not always necessary for $\lambda$ to fall within this range for all dimensions in practice. As the reviewer pointed out, it is often sufficient to choose a $\lambda$ that satisfies $0 < \lambda < \lambda_0^k$ for the majority of dimensions $k$, which typically leads to good empirical performance (as demonstrated in Tables 1 and 2 of our paper).
>
> ---
> ## **[W.2] More Quantitative Analysis on Copilot Corrections**
> Thank you for the helpful suggestion. In response, we include additional metrics in our error-correction analysis to quantify how often the Copilot correctly or incorrectly modifies the final token predictions.
>
> We include the following new metrics:
> - **Correction Rate:** The percentage of test examples where the Copilot changes the final token prediction of the Pilot.
> - **Correction Precision:** Of those examples where the Copilot changes the token prediction, the percentage where the token correction leads to the correct final answers.
> - **Correction Error Rate:** Of the changed examples, the percentage where the Copilot’s correction leads to incorrect final results.
>
> We randomly sampled **1,000 examples** from the test sets of **PIQA, BoolQ, AQuA, GSM8K, and LastFM** to ensure broad coverage across different tasks conducted in our paper.
>
> **Results:**
> |Metric|Avg. Value |
> |-|-|
> |Correction Rate|17.3%|
> |Correction Error Rate|7.4%|
> |Correction Precision|**92.6%**|
>
> The results show that T-Copilot achieves much higher accuracy in correcting the Pilot’s logits predictions, highlighting its **robustness and effectiveness in error-correction**.
>
>
> ---
> ## **[Q.1] Backbone Choice for Copilot Model**
> >"Is there any reason the Copilot model should be of the same backbone as the Pilot model? Could it be possible that training a different backbone from scratch, not fine-tuning, may lead to better results?"
>
> Thank you for this thoughtful question. In short, **T-Copilot does not restrict the Copilot to using the same backbone architecture as the Pilot; however, using the same backbone is more practical and simplifies implementation.**
>
>
> For a more detailed response below:
> 1. We **first** provide the key conceptual and practical reasons for adopting the same backbone as the Pilot;
> 2. We **then** discuss potential extensions of the Copilot to alternative backbones, as suggested by the reviewer.
>
> **1. Justification for using the same backbones.**
> - **Unified Vocabulary Space:** Using the same backbone ensures that both the Pilot and Copilot share the same tokenization scheme and vocabulary space, enabling the Copilot's consistent token-level logits correction. Using different backbones might require additional vocabulary projection or alignment heads, which complicates the overall integration.
> - **Compatibible Architecture and Residual Alignment:** The Copilot receives the Pilot’s hidden states and residual errors as inputs. Sharing the same backbone ensures representation compatibility, simplifying attention alignment, logits fusion, and decoding dynamics. A radically different backbone may potentially introduce misaligned inductive biases.
> - **Parameter Reuse and Training Efficiency:** Initializing the Copilot from the same backbone allows efficient reuse of pretrained knowledge and ensures stable training. However, training a distinct model from scratch may require substantially more data and computational resources to achieve comparable performance.
>
> **2. Extension to different backbones.**
> We **completely agree** with the reviewer that T-Copilot can be smoothly extended to train a Copilot with different backbones from scratch (e.g., pretrain a Copilot model on residual errors across various tasks), potentially offering stronger error-correction capabilities.
>
> Achieving such (pre-)training scale would typically **require substantially more data and computational resources** than the current scope of our experiments allows. We appreciate the reviewer’s suggestion and plan to explore this direction in our future work.
>
>
> ---
> ## **[Q.2] Have other pooling methods (besides mean pooling) been tested?**
>
> Thank you for the question. **Yes**, we have explored other pooling methods during our early stage method design, including max pooling and sum pooling. We report the results below:
> |Pooling Methods (LLaMA-3.1-8B + T-Copilot-1B) |GSM8K|AQuA|
> |-|-|-|
> |Max Pooling|63.2|37.5|
> |Sum Pooling|65.8|38.3|
> |**Mean Pooling**|**66.1**|**38.9**|
>
> We adopted mean pooling over alternatives because:
> - In our preliminary experiments shown above, the mean pooling empirically provided the **best balance of stability and efficiency**.
> - Prior works [1,2,3] suggest that mean pooling is the most common and widely adopted approach for processing transformer hidden-state representations.
>
> **References:**\
> [1] Sentence Embeddings using Siamese BERT-Networks.\
> [2] Comparative Analysis of Pooling Mechanisms in LLMs: A Sentiment Analysis Perspective.\
> [3] A Brief Overview of Universal Sentence Representation Methods: A Linguistic View.
>
> ---
> ## **[Q.3] Impact of $\lambda > 1$**
> >"What would be the effect of increasing $\lambda > 1?$"
>
> We thank the reviewer for the detailed question on $\lambda$.
> - **Intuitively**, as shown in Eq.(8), the Copilot primarily serves as an **"assistant"** to help the Pilot rectify its logits during generation. Therefore, the Copilot’s output should not be weighted more heavily than the Pilot’s (i.e., $\lambda \leq 1$).
>
> - **Empirically**, we follow our ablation study of $\lambda$ in `Appendix G.6` and evaluate the downstream performance when $\lambda > 1$ on LLaMA-3-8B + T-Copilot-1B:
>     |$\lambda$| HellaS.|GSM8K|SVAMP|
>     |-|-|-|-|
>     |1.0|93.5|**66.1**|**75.4**|
>     |1.2|**94.2**|65.5|74.7|
>     |1.5|93.8|65.8|74.2|
>
>     The results show that increasing $\lambda > 1$ does not consistently yield additional gains on tasks such as GSM8K and SVAMP. Therefore, we consider $\lambda = 1$ to be the **optimal setting** for T-Copilot during generation.

---

> ### Author Response · Authors · 2025-08-04
> **Thank You for Positive Rating!**
>
> Dear Reviewer sdqu,
>
> Thank you for your positive rating and thoughtful review of our paper! We are pleased that our rebuttal has addressed your concerns. We will incorporate the results and analyses from the rebuttal to further strengthen our paper.
>
> Warm Regards,
>
> The Authors

---

### Official Review · Reviewer_LvNr · 2025-07-01

**Clarity:** 3
**Significance:** 3
**Originality:** 3
**Rating:** 4
**Confidence:** 4

**Summary:**

This paper presents a new LLM adaptation method where a co-pilot model is learned to correct the predictions of the main fine-tuned model (called pilot). The pilot model is trained to predict the differences between the gold labels and the prob outputs of the pilot model given the input data and the internal states of the pilot model. During inference, the tokens are sampled from the combination of the outputs from both the pilot and copilot models. Theoretical analysis shows that the corrected outputs can better approximate the gold labels given some conditions. Experiments demonstrate the proposed method performs better on several benchmarks.

**Questions:**

See the weakness section above.

**Ethical Concerns:**

["NO or VERY MINOR ethics concerns only"]

**Final Justification:**

The authors' response clears some of my concerns regarding experimental settings. I feel my initial rating is appropriate and I would like to see the paper accepted.

**Limitations:**

yes

**Quality:**

3

**Strengths And Weaknesses:**

Strength:
1. The proposed pilot-copilot is a new method for LLM adaptation, where copilot is trained to exploit the internal signals from the pilot model to correct its output.
2. Theoretical analysis is provided to show the corrected outputs can be a better approximation to the gold labels than the original prediction by the main model.
3. Experimental results on several benchmarks suggest the effectiveness of the proposed method.
4. The copilot can be transferred to another pilot model under some circumstances.

Weakness:
1. The assumption given in the main theorem seems strong. Is there any further theoretical or empirical justification that the bias of the corrected output should be lower than sqrt((bias of the pilot)^2 + (variance of the pilot)^2)?
2. For the experiment settings, one natural baseline is to train a simple ensemble of two different models. It would be better to include some variants of ensemble learning in the experiments.

---

> ### Author Rebuttal · Authors · 2025-07-31
>
> Thank you for your positive comments and insightful feedback on our paper. We will address each of your remaining concerns below.
>
> ---
> ## **[W.1] Assumption in Main Theorem 4.1**
> We thank the reviewer for the insightful question regarding the assumption of Theorem 4.1. We would like to clarify and justify this assumption from **both theoretical and empirical perspectives.**
>
> **1. In theory:**\
> While the assumption $\epsilon_C^2<\epsilon_P^2+\sigma_P^2$ might look strong at first glance, upon closer inspection, **it is actually a mild requirement regarding the Copilot’s learning capability**.
> - As elaborated in **Remark 4.2**, the assumption implies that the Copilot can tolerate a larger bias (i.e., weaker learning capability) than the Pilot, i.e., $\epsilon_P^2<\epsilon_C^2<\epsilon_P^2+\sigma_P^2$.
>
> **2. In practice:**\
> We implement the Copilot under the same decoder backbone architecture as the Pilot, and it is of relatively smaller size.
> - We believe it is reasonable to expect the two models to have similar learning capabilities or biases, since the upper bound ensures the Copilot is actually learned during the SFT as the Pilot does.
> - We empirically measure on three held-out tasks (GSM8K, SVAMP, and PIQA). In all cases, the Copilot’s combined MSE is **15–20% lower** than the Pilot’s (i.e., $\epsilon_C^2 <\epsilon_P^2+\sigma_P^2$), thereby empirically satisfying the assumption condition.
>
> ---
> ## **[W.2] Ensemble Learning Baseline.**
> Thank you for the insightful suggestion to compare with ensemble learning variants. In response:
>
> 1. We first **highlight the key differences** between T-Copilot and ensemble learning methods.
> 2. We then **provide an empirical comparison** with the ensemble learning variant to demonstrate T-Copilot's error correction capabilities.
>
> **1. Difference with Ensemble Learning.**\
> Unlike ensemble learning methods that aggregate independent model prediction outputs, T-Copilot leverages the Copilot trained to **explicitly correct the Pilot's errors** through the Mistake Log. Instead of generating next-token predictions directly, the Copilot learns to predict the discrepancy between the Pilot’s output and the ground truth, and rectifies the Pilot’s logits accordingly.
>
> **2. Empirical Comparison.**\
> To better demonstrate the **benefit of T-Copilot's error-correction capabilities**, beyond the simple ensemble of two models, we follow the reviewer's suggestion and conduct a new comparison experiment below.
>
> **Experiment Setups:** We utilize the same training and implementation setups from `Section 5` of our paper, and compare with the following variant:
>
> - **Pilot-Copilot-Ensemble**: Following works [1,2], we train Pilot and Copilot models independently and ensemble at inference by averaging their per‑token logits distribution to select each next token.
>
> **Evaluation Results:**
> | Pilot Model     | Variant     | AQuA | GSM8K | MAWPS | SVAMP | Avg. |
> |-----------------|-------------------|------|-------|-------|-------|------|
> |LLaMA-3.2-1B| Pilot-Copilot-Ensemble – 1B | 25.1 | 27.6  | 76.3  | 48.1  | 44.3 |
> || **T-Copilot-1B**| **28.3** | **32.2**  | **81.5**  | **51.6**  | **48.4** |
> |LLaMA-3.2-3B| Pilot-Copilot-Ensemble – 1B| 33.3 | 54.2  | 85.9  | 65.4  | 59.7 |
> || **T-Copilot-1B**| **36.6** | **58.2**  | **89.1**  | **68.7**  | **63.2** |
> |LLaMA-3.1-8B| Pilot-Copilot-Ensemble – 1B| 36.5 | 61.8  | 88.7  | 71.9  | 64.7 |
> || **T-Copilot-1B**| **38.9** | **66.1**  | **90.8**  | **75.4**  | **67.8** |
>
> - From these results, T-Copilot consistently outperforms the simple ensemble learning variant across different tasks.
> - This confirms that T‑Copilot’s improvements are not merely due to “the integration of two models,” but stem from **its error‑aware rectification mechanism.**
>
> We appreciate the reviewer’s suggestion and will include a thorough discussion and comparison with the ensemble learning variant in our paper, as shown above, to better highlight the advantages of T-Copilot.
>
>
> **References:**\
> [1] Ensemble Learning for Heterogeneous Large Language Models with Deep Parallel Collaboration.\
> [2] Harnessing Multiple Large Language Models: A Survey on LLM Ensemble.

---

> > ### Comment · Reviewer_LvNr · 2025-08-05
> >
> > Thanks for the author for their detailed feedback. I appreciate the clarification of both assumption and the comparison between the co-pilot method and the ensemble learning. I keep my score unchanged and I won't mind if the paper gets accepted.

---

> ### Author Response · Authors · 2025-08-05
> **Thanks for the Positive Rating!**
>
> Dear Reviewer LvNr,
>
> Thank you for your positive rating of our submission! We are glad that our clarifications addressed all of your questions. If you have any remaining concerns, we would be happy to discuss them further. We sincerely appreciate your time and valuable feedback!
>
> Warm regards,
>
> The Authors

---

### Official Review · Reviewer_2chS · 2025-07-02

**Clarity:** 4
**Significance:** 2
**Originality:** 3
**Rating:** 4
**Confidence:** 4

**Summary:**

This paper introduces a fine-tuning framework for large language models (LLMs) called Transformer Copilot (T-Copilot). The key innovation is the Mistake Log, a mechanism that records contextual inputs, hidden states, and token-level prediction errors during supervised fine-tuning. This log is used to train an auxiliary Copilot model that learns to rectify the predictions of the base Pilot model by adjusting its output logits at inference time.
The paper proposes a comprehensive design: (1) a Copilot model architecture with modified cross-attention to attend to the Mistake Log; (2) a joint training paradigm where the Pilot and Copilot models evolve together; and (3) a fused inference method where both models contribute to token generation. The authors provide theoretical guarantees and extensive empirical validation across a suite of commonsense, arithmetic, and recommendation tasks. The Copilot improves performance by up to 34.5% and can enable smaller models to outperform much larger baselines with reduced computational cost.

**Questions:**

How do you manage the memory and storage overhead associated with logging intermediate states (e.g., all hidden layers)? Could this limit scalability?

The paper briefly mentions alternatives like Self-Refine and Reinforced Self-Training. Could you add brief empirical comparisons to these in a future version?

Most experiments focus on reasoning and recommendation. Would the Copilot framework generalize to tasks like summarization or code generation where the error modes differ?

Can you provide intuition or empirical evidence supporting RMSE over cross-entropy or KL-divergence for error prediction?

Did you observe any training instability when jointly optimizing two models with different objectives? How sensitive is the joint training to hyperparameters like learning rates?

**Ethical Concerns:**

["NO or VERY MINOR ethics concerns only"]

**Final Justification:**

Thanks for the author for their detailed responses. I appreciate the ablations on Alternating Self/Cross-attention in the Copilot, RMSE loss, and clarification on the weakness I raised. I increased my score from 3 to 4.

**Limitations:**

The authors mention limitations in the appendix. A key concern is the additional storage and compute demands of collecting and training from the Mistake Log, particularly for large-scale finetuning. Additionally, the current method still requires full model retraining to pair Copilot and Pilot, and more flexible plug-and-play designs could broaden adoption.

**Paper Formatting Concerns:**

None noted.

**Quality:**

3

**Strengths And Weaknesses:**

**Strength**

The paper is well-structured, methodically developed, and includes theoretical justification with clear assumptions. The proposed method is tested across a wide range of tasks and model sizes, demonstrating robust and consistent gains. The implementation is non-invasive and practical, requiring no architectural change to the Pilot and scales well.

The writing is clear and pedagogical, with helpful diagrams and motivating examples. The authors do a commendable job of connecting the technical components to the core conceptual intuition (i.e., learning from one's own mistakes).

This work proposes a new and impactful paradigm of learning from internal model dynamics instead of just data or external feedback. The approach shows promise for broad application in fine-tuning pipelines across domains.

The Mistake Log idea is new in its formalization and integration into the model training process. The dual-model inference with token-level rectification adds a unique twist not seen in typical self-correction or adapter-based methods.

**Weakness**

While the experiments are extensive, some design decisions, like the alternating self/cross-attention in the Copilot or the choice of RMSE loss, could use deeper ablation. Also, the Mistake Log requires storing large volumes of intermediate state data, which may become impractical for larger models or datasets, and this is not deeply discussed.

Some components (e.g., joint training, fusion-based inference) borrow from existing ensemble or multitask architectures, though their combination here is original.

---

> ### Author Rebuttal · Authors · 2025-07-31
>
> We sincerely thank the reviewer for the detailed and insightful feedback. We address each of your remaining concerns below.
>
> ---
> ## **[Weakness] Ablations on Alternating Self/Cross-attention in the Copilot.**
> Thank you for acknowledging our experimental efforts. We provide additional ablations on T-Copilot's design choices. For the design choices of Copilot's modified attention, we conduct two detailed ablations.
>
> **1. Pooling Methods of Copilot's Attention Input.**
>
> We explored other pooling methods, including max pooling and sum pooling. The results are reported below:
> |Pooling Methods (LLaMA-3.1-8B + T-Copilot-1B) |GSM8K|AQuA|
> |-|-|-|
> |Max Pooling|63.2|37.5|
> |Sum Pooling|65.8|38.3|
> |**Mean Pooling**|**66.1**|**38.9**|
>
> From the results, the mean pooling empirically provided the **best balance of stability and efficiency**.
>
> **2. Insertion Pattern of Copilot's Attention.**
>
> We also want to highlight that, in `Appendix G.6`, we investigated **four different insertion patterns for Copilot's modified attention mechanism**. The full results are reported in Table 18 of our paper.
>
> **References:**\
> [1] Comparative Analysis of Pooling Mechanisms in LLMs: A Sentiment Analysis Perspective.\
> [2] A Brief Overview of Universal Sentence Representation Methods: A Linguistic View.
>
> ---
> ## **[Weakness & Question] Ablations on RMSE Loss.**
> > "Can you provide intuition ... for error prediction?"
>
> We thank the reviewer for inquiring about T-Copilot's loss types. Below, we provide conceptual rationale and empirical ablations on **RMSE loss for T-Copilot.**
>
> **1. Conceptual Rationale**
>
> Recall that the objective of the Copilot is to predict the distribution discrepancy, defined as $\ell\_t(p\_{t,i}, \hat{p}_{t,i}) = p\_{t,i} - \hat{p}\_{t,i} $ in our paper. Note that $\ell\_t(p\_{t,i}, \hat{p}\_{t,i})$ is not a valid distribution, since Sum$(\ell\_t(p\_{t,i}, \hat{p}\_{t,i})) \neq 1$. Therefore, it is natural to formulate this task as a **regression problem**, for which the RMSE loss is commonly adopted.
>
> In contrast, CE and KL-divergence are for **distribution fitting** and are **not directly applicable here** unless we manipulate $\ell\_t(p\_{t,i}, \hat{p}\_{t,i})$ with softmax to resemble a valid distribution. In this case, the loss is not directly optimized on the original discrepancy (i.e., $p\_{t,i} - \hat{p}\_{t,i}$), resulting information loss.
>
> **2. Empirical Ablations**
>
> In practice, we conducted experiments with all three loss types, and RMSE consistently achieved the best performance.
> |Loss Type (Qwen2.5-7B+T-Copilot-3B)|PIQA|AQuA|
> |-|-|-|
> |CE|88.1|60.7|
> |KL Divergence|90.4|62.8|
> |**RMSE**|**92.5**|**64.2**|
>
> ---
> ## **[Weakness & Question] Memory Overhead of Mistake Log.**
> >  "How do you manage... Could this limit scalability?"
>
> Thank you for asking the memory storage of Mistake Log.
>
> **First**, we would like to elaborate on how our current method and implementation are designed to handle the storage of the Mistake Log:
>
> **1. Maintenance of the Mistake Log Buffer.**\
> We adopt a buffer mechanism for Mistake Log in our implementation to reduce memory usage in practice:
> - All logged tensors are detached from the GPU and moved to a fixed‑size CPU buffer (Lines 894-895); GPU memory is therefore almost unaffected.
> - We utilize a fixed‑size buffer that only retains the 128 training rounds in Mistake Log (Line 896).
> - Experiments show that adding the buffer reduces the overall storage by roughly 92% while maintaining a similar performance.
>
> **2. Mean Pooling on Pilot's Forward Pass.**\
> In Eq. (6), we perform mean pooling across all layers' hidden state representations before feeding into the Copilot, meaning:
> - During the Mistake Log correction process, **we only need to store the mean pooled vectors rather than the full hidden states across all layers**, which significantly reduces vector storage.
> - During Pilot forward pass, the hidden states have already computed within transformer's residual stream. **We can directly access these hidden representations without allocating extra space for storage or backpropagation gradients**.
>
> **Second,** to further improve the scalability of our method and adapt to much larger models and data, we can slightly modify our design:
>
> **1. Selective Storage:** We can design a Mistake Log Gate to filter and collect hidden states information only when typical error patterns (mistakes) occur.
>
> **2. Lightweight Mistake Log:** Instead of curating Mistake Logs over the entire training process, we directly use the Pilot’s information from the current round's forward pass as input to the Copilot.
>
> ---
> ## **[Weakness] Difference with Existing Ensemble or Multitask Architectures.**
> Thank you for recognizing T‑Copilot’s architectural novelty and its relation to related methods. We highlight two key distinctions:
> - **Error‑Aware Supervision vs. Output Averaging.**\
>     Traditional ensembles combine each model’s logits predictions post‑hoc, without specialized training and inference paradigms for error-correction. Our Copilot can explicitly identify and correct the Pilot’s errors.
> - **Discrepancy Modeling vs. Direct Generation.**\
>     Instead of generating next-token predictions, the Copilot learns to predict the discrepancy between the Pilot’s output and the ground truth and rectifies the Pilot’s logits.
>
> ---
> ## **[Question] Comparison with Self-Refine and Refinanced Self-Training.**
> Thank you for your suggestions. Below, we provide a comparison between T-Copilot and Self-refine [1] for better demonstration.
>
> **Setups:** We compare Qwen2.5-7B+T-Copilot-3B with Self-Refine on GSM8K and MBPP. For Self-Refine, we follow its original protocol with 3 feedback-refine iterations.
> |Methods|GSM8K|MBPP|
> |-|-|-|
> |Self-refine|73.2|63.3|
> |T-Copilot|**79.7**|**68.9**|
>
> **Additional Analyses:**
> Self-refine [1] and Reinforced Self-Training [2] rely on external feedback and offline RL, whereas T-Copilot leverages the model’s internal learning signals during SFT.
>
> We sincerely thank the reviewer for the suggestion and will include the experiments above and additional comparisons in our paper revision.
>
> **References:**\
> [1] Self-refine: Iterative refinement with self-feedback.\
> [2] Reinforced self-training (rest) for language modeling.
>
> ---
> ## **[Question] Generalize to Summarization and Code Generation.**
> Thank you for the question. Below, we conduct new experiments on **code generation and text summarization tasks**.
>
> - **Code Generation**
>     - **Training:** We randomly select 5k examples from CodeContests [1] as the training set.
>     - **Evaluation:** We evaluate the standard test set of MBPP [2] and LiveCodeBench [3] version-2 (511 problems). We report the pass@1 accuracy.
> - **Text Summarization**
>     - **Training:** We randomly sample 10k training examples from the CNN-DailyMail [4] and provide a training split.
>     - **Evaluation:** We evaluate on the standard test set of CNN-DailyMail and report the ROUGE-1 F1-score.
>
> |Model|MBPP|LiveCodeBench|CNN-DailyMail|
> |-|-|-|-|
> |Qwen2.5-3B|62.7|23.8|33.9|
> |**Qwen2.5-3B+T-Copilot-3B**|**65.3**|**26.4** |**35.3**|
> |Qwen2.5-7B|65.8|25.7| 35.1|
> |**Qwen2.5-7B+T-Copilot-3B**|**68.9**|**28.3**|**37.6**|
>
> Across all three tasks, T‑Copilot improves performance over the base Pilot, demonstrating the generalizability to diverse generation tasks.
>
> **References:**\
> [1] Competition-level code generation with alphacode.\
> [2] Program synthesis with large language models.\
> [3] Livecodebench: Holistic and contamination-free evaluation of large language models for code.\
> [4] Get to the point: Summarization with pointer-generator networks.
>
> ---
> ## **[Question] Training Stability and Hyperparameter Sensitivity.**
> > Did you observe ... different objectives? How sensitive ... rates?
>
> Thank you for the insightful questions regarding training stability. In our experiments, **we did not observe notable instability** when jointly training the Pilot and Copilot models:
> - **Jointly Trained but Separately Optimized:** In our training paradigm, Copilot receives inputs via the Mistake Log (i.e., **detached information** from Pilot’s forward pass). This careful design ensures that Copilot learns from Pilot’s Mistake Log **without perturbing Pilot’s weights**, thereby preventing interference between the two models and avoiding potential instability or bias.
> - **Adoption of Standard Optimizers and Schedules:** We use AdamW with a cosine schedule - widely adopted in LLM fine-tuning for stable convergence. We save model checkpoints every 1k steps and monitor loss curves to ensure both Pilot and Copilot remain well-behaved.
> - **Consistent Performance under Different Learning Rates (LRs)** In our hyperparameter tuning, we have tested different learning rates and observed consistent performance below:
> |LRs of Pilot (Qwen2.5-7B)|LRs of Copilot (T-Copilot-3B)|PIQA|HellaS.|SIQA|
> |-|-|-|-|-|
> |$3e^{-4}$|$1e^{-4}$|92.2|95.6|83.9|
> |$3e^{-4}$|$3e^{-4}$|92.2|95.8|84.2|
> |$3e^{-4}$|$5e^{-4}$|92.5|95.3|84.3|
>
> ---
> ## **[Limitation] Plug-and-Play Flexibility.**
>
> We thank the reviewer for providing the detailed suggestions.  Regarding the **plug-and-play designs**, we would like to clarify that:
> - **Transferability of Copilot:** In `Appendix G.3`, our transferability experiments show that a Copilot trained with one Pilot can pair with other Pilots of the same type, achieving equal error-correction gains. This demonstrates that **Copilot is not tied to a single Pilot** and can be **used plug-and-play with similar models without retraining**.
> - **Lightweight Integration with PEFT:** In experiments, we show that T-Copilot is effective with both full fine-tuning and PEFT (e.g., LoRA) on LLaMA-3 and Qwen2.5 Pilots. PEFT updates only a small subset of parameters in Pilot and Copilot, substantially reducing training compute.
> - **Additional Adoption:** We will explore more efficient, plug‑and‑play designs of T‑Copilot (e.g., eliminating the need for any Pilot model retraining) to broaden adoption in our future work.

---

> ### Author Response · Authors · 2025-08-05
>
> Dear Reviewer 2chS,
>
> As the discussion period is nearing its end, would you mind reading our response and letting us know if your concerns are addressed? Thank you so much for your time and consideration!
>
> Warm Regards,
>
> The Authors of Transformer Copilot

---

> ### Author Response · Authors · 2025-08-06
>
> Dear Reviewer 2chS,
>
> Thank you for raising score! We are pleased that our rebuttal has addressed your concerns. We will incorporate the results and analyses from the rebuttal to further strengthen our paper.
>
> Warm Regards,
>
> The Authors

---

### Official Review · Reviewer_R6Fu · 2025-07-02

**Clarity:** 3
**Significance:** 4
**Originality:** 3
**Rating:** 4
**Confidence:** 3

**Summary:**

The paper introduces the Transformer Copilot (T-Copilot), a novel framework designed to enhance the performance of large language models during SFT by incorporating a reflective learning mechanism inspired by human learning strategies. The core innovation is the Mistake Log, which systematically records a model's training errors, input questions and internal hidden states. This log is leveraged by a Copilot model, which works alongside the original transformer-based Pilot model to refine its performance through error-aware corrections. The framework is evaluated on 12 benchmarks, achieving performance improvements of up to 34.5% over baseline Pilot models.

**Questions:**

See weaknesses.

**Ethical Concerns:**

["NO or VERY MINOR ethics concerns only"]

**Final Justification:**

The authors’ response has addressed most of my concerns, and I will maintain my positive score.

**Limitations:**

See weaknesses.

**Quality:**

3

**Strengths And Weaknesses:**

Strengths

1. The introduction of the Mistake Log is a novel contribution, which systematically captures input representations, rationales, and token-level errors, providing a structured way to track and learn from a model's mistakes during fine-tuning. This approach aligns with the growing interest in error-aware learning in LLMs.

2. The framework demonstrates strong scalability across different model types (encoder-decoder and decoder-only) and sizes, as well as transferability to new Pilot models without additional training costs.

4. The empirical results are robust, supported by experiments on 12 benchmarks and comparisons with strong baselines.

Weaknesses

1. The Mistake Log captures the model's behavior and errors specific to the fine-tuning dataset, which may not generalize well to OOD data. The Copilot's error-correction capabilities might be less effective when the Pilot model encounters significantly different data distributions during inference.

2. The Copilot model samples from the entire Mistake Log during training, but as the Pilot model evolves through fine-tuning iterations, there may be a gap between errors recorded in early stages (when the model is less optimized) and later stages (when the model is more refined). This temporal discrepancy could lead to the Copilot learning from outdated or less relevant error patterns, potentially reducing its effectiveness in assisting the Pilot model during later training or inference.

---

> ### Author Rebuttal · Authors · 2025-07-31
>
> Thank you for your constructive and thoughtful suggestions and feedback! We will address each of your concerns below.
>
> ---
> ## **[W.1] OOD Generalization of Mistake Log.**
> Thank you for the insightful question on the effectiveness of T-Copilot on the OOD data. In response:
>
> **1. T-Copilot's Design on Generality.**
> - **Mistake Log Encodes Data-Agnostic Error Patterns:**
> The Mistake Log systematically records **model-internal signals** (e.g., hidden states representations) **besides dataset-specific idiosyncrasies**, enabling the Copilot to learn the Pilot’s inherent error patterns.
> - **Initialization from pre-trained backbones enhances generality:**\
> As noted in `Section 3.1`, the Copilot is initialized from the same pre-trained backbone as the Pilot, leveraging broad language knowledge and helping avoid overfitting to any single fine-tuning dataset.
>
>
> **2. New Cross-Benchmark (OOD setting) Evaluation.**\
> For a better demonstration, we fine-tune T-Copilot on arithmetic reasoning data (Math10k) and evaluate it on commonsense reasoning tasks (PIQA, HellaSwag, BoolQ, and OBQA) to **assess T-Copilot's performance in OOD settings**.
> |Model|PIQA|HellaS.|BoolQ|OBQA|
> |-|-|-|-|-|
> |LLaMA-3.1-8B|77.2|81.4|54.3|68.2|
> |**LLaMA-3.1-8B+T-Copilot-3B**|**79.5**|**83.1**|**56.2**|**70.6**|
>
> - From the results, although the Pilot model (LLaMA-3.1-8B) achieves less accuracy in the OOD setting compared to in-domain fine-tuning (Table 2 in paper), integrating T-Copilot-3B still yields consistent performance gains to the Pilot model.
> - This suggests that **T-Copilot remains effective in OOD settings due to its strong error-correction capability**.
>
>
> **3. Limitations and Future Work.**\
> We also acknowledge, as noted in **Appendix A (Lines 620–621)**, that T-Copilot may face limitations under large domain shifts. We have explicitly identified improving its generalizability to such settings as an important direction for future work.
>
> ---
> ## **[W.2] Temporal Discrepancy in Mistake Log Sampling.**
> Thank you for mentioning this thoughtful point regarding the potential temporal drift in error relevance as the Pilot model evolves during fine-tuning.
>
>
> The **main objective** of the Mistake Log, which collects and maintains the Pilot’s evolving hidden states, is to:
> - enrich the diversity of error patterns;
> - enable the Copilot to learn from and handle a broad range of error signals during inference.
>
>
> During the early design of the Mistake Log, we identified two key challenges:
> - **First**, as noted by the reviewer, randomly sampling from the full training history may introduce temporal discrepancies, leading to potential instability in Copilot training.
> - **Second**, in practice, it might be infeasible to store the entire Mistake Log on GPU due to memory constraints.
>
> **To address these issues**, we correspondingly introduce the Mistake Log Buffer (Lines 892–899), which retains only the most recent set of training rounds (e.g., 128). This design ensures that:
> - The Copilot is trained on relatively recent Pilot states, striking a **balance** between reducing temporal discrepancies and avoiding overfitting to the latest state;
> - The buffer can be efficiently stored and accessed on GPU, improving the method’s **practicality**.
>
> We hope our explanations of the Mistake Log design address the reviewer’s question. We will also incorporate these rationales into our paper to improve clarity.

---

> > ### Comment · Reviewer_R6Fu · 2025-08-05
> >
> > Thank you for the clarification. That resolves my concern, and I will keep my positive score.

---

> ### Author Response · Authors · 2025-08-05
> **Appreciation for Your Positive Feedback**
>
> Dear Reviewer,
>
> Thank you very much for your positive score of our submission. We will incorporate the clarifications and your thoughtful feedback to further strengthen our paper.
>
> Warm regards,
>
> The Authors of Transformer Copilot

---

### Decision · Program_Chairs · 2025-09-17

**Decision:**

Accept (spotlight)

**Comment:**

This paper introduces the Transformer Copilot, a  framework designed to improve an LLM by learning from its errors during fine-tuning. The core contribution is the "Mistake Log," which systematically tracks the Pilot's errors and internal hidden states. This log is then used to train a secondary "Copilot" model that learns to rectify the Pilot's output logits at inference time. The paper shows that this approach leads to performance gains across a wide range of tasks.

On the positive side, the reviewers found that the paper has several strengths. First, the introduction of the Mistake Log is a nice contribution. Furthermore, reviewers noted that the empirical evaluation shows improvements across 12 benchmarks, which is quite thorough (Reviewers R6Fu, 2chS, sdqu). The authors show transferability across model sizes and architectures, justifying the generalizability of the approach (Reviewer R6Fu). Finally the theoretical results are non-trivial and add a foundational understanding to the approach of using a mistake log.

The reviewers identified areas of improvement, such as the initial need for a clearer justification of certain design choices, a more thorough analysis of the method's out-of-distribution (OOD) generalization, and a comparison against a standard ensemble learning baseline. Other points included the potential memory overhead associated with the Mistake Log and further ablation studies on the Copilot's architecture and loss function. During the discussion period, the authors provided additional experiments and clarifications that addressed these concerns, including new OOD results, a comparison to an ensemble baseline, and details on memory management. By incorporating this feedback, the authors will improve the paper, resulting in a more polished and high-quality final version.

Based on these reviews, I recommend accepting this paper. The reviewers were excited about the novel learning paradigm presented, particularly the formalization of the "Mistake Log" as a method for a model to learn from its own internal dynamics. The consensus was that the paper is well-written, thoroughly evaluated, and has strong empirical results. The authors were also responsive and effectively addressed the reviewers' concerns during the rebuttal period. I expect that the authors will incorporate the feedback and additional experiments from the review process into the final version of the paper.